**Data Availability Statement:** Northwestern Arch: https://doi.org/10.21985/n2-8hxs-md53.

**Funding:** National Institute of Neurological Disorders and Stroke, T32 NS047987, Melisa Menceloglu.

# A phase-shifting anterior-posterior network organizes global phase relations

**Melisa Menceloglu[1]¤, Marcia Grabowecky[1,2], Satoru Suzuki[1,2]***

**1** Department of Psychology, Northwestern University, Evanston, Illinois, United States of America,
**2** Interdepartmental Neuroscience, Northwestern University, Evanston, Illinois, United States of America

¤ Current address: Department of Psychology, Michigan State University, East Lansing, Michigan, United States of America
* satoru@northwestern.edu

## Abstract

Prior research has identified a variety of task-dependent networks that form through inter-regional phase-locking of oscillatory activity that are neural correlates of specific behaviors. Despite ample knowledge of task-specific functional networks, general rules governing global phase relations have not been investigated. To discover such general rules, we focused on phase modularity, measured as the degree to which global phase relations in EEG comprised distinct synchronized clusters interacting with one another at large phase lags. Synchronized clusters were detected with a standard community-detection algorithm, and the degree of phase modularity was quantified by the index $q$. Notably, we found that the mechanism controlling phase modularity is remarkably simple. A network comprising anterior-posterior long-distance connectivity coherently shifted phase relations from low-angles ($|\Delta\theta| < \pi/4$) in low-modularity states (bottom 5% in $q$) to high-angles ($|\Delta\theta| > 3\pi/4$) in high-modularity states (top 5% in $q$), accounting for fluctuations in phase modularity. This anterior-posterior network may play a fundamental functional role as (1) it controls phase modularity across a broad range of frequencies (3–50 Hz examined) in different behavioral conditions (resting with the eyes closed or watching a silent nature video) and (2) neural interactions (measured as power correlations) in beta-to-gamma bands were consistently elevated in high-modularity states. These results may motivate future investigations into the functional roles of phase modularity as well as the anterior-posterior network that controls it.

## Introduction

While white-matter neural connections are relatively fixed, on-demand functional neural networks may form through coordinated oscillations of neural excitability, selectively facilitating communications among task-relevant neural populations [1, 2]. Such functional networks at a coarse spatial scale have been non-invasively investigated in humans by examining phase relations in the sinusoidal components of EEG/MEG signals between scalp sensors (or derived sources). "Functional connectivity" is inferred between a pair of sensors (sources) when their phase difference, $\Delta\theta$, becomes stable. Researchers have primarily focused on $\Delta\theta$ stability in the mid-range (e.g., by focusing on the imaginary component of $\Delta\theta$ [3–5] or orthogonalizing the

**Competing interests:** The authors have declared that no competing interests exist.

to-be-compared sinusoidal signals [6, 7]) as $|\Delta\theta|\sim0$ could reflect volume-conduction effects and $|\Delta\theta|\sim\pi$ could reflect a single dipole rather than the oscillatory coordination of neural excitability between separate neural populations. This line of investigation, often using graph theoretic measures to characterize functional connectivity structures, has identified numerous within- and cross-frequency functional networks underlying a variety of perceptual and cognitive operations ([8–12]; for reviews see [13–16]). Phase coherence measures have also been used to examine global functional networks which have been compared with networks derived with DTI [17] or fMRI BOLD correlations [18]. While prior studies primarily focused on the strength of phase coherence to infer functional connectivity, the current study undertook a complementary effort to investigate general rules governing global phase relations to understand the general context in which on-demand phase realignments occur. Whereas phase relations (lags) have been examined to infer directionality of neural communications between specific regions [19, 20 for a review], no prior studies (to our knowledge) have systematically examined the global topology and dynamics of phase relations in human EEG.

What would be an effective way to characterize a global organization of phase relations? We reasoned that a highly organized state may be characterized by high phase modularity, comprising clusters of synchronously oscillating populations ($|\Delta\theta|\sim0$) interacting with one another at large phase lags ($|\Delta\theta|\gg0$). In other words, in a highly organized state, within-cluster inter-site $|\Delta\theta|$ would be low while between-cluster inter-site $|\Delta\theta|$ would be high, whereas in a minimally organized state, within-cluster and between-cluster inter-site $|\Delta\theta|$ would be similar. To quantify such phase modularity, we used a standard community-detection algorithm [14, 21]. We used the index, $q$, referred to as maximized modularity (varying between 0 and 1), quantifying the degree to which within-cluster $|\Delta\theta|$'s were lower than between-cluster $|\Delta\theta|$'s, computed at each time point (per frequency per condition per participant). The MATLAB code of the algorithm we used to compute $q$ based on the eigenvectors of modularity matrices is provided in S1 File (modularity_und.m by Rubinov, Power, Bassett, Wang, and LaPlante, 2017).

Temporal distributions of $q$ were approximately normal and virtually invariant across the representative frequencies (ranging from 3 to 50 Hz) and the behavioral conditions (resting with the eyes closed, resting with the eyes open in a dark room, or watching a silent nature video) (Fig 1). This suggest that similar mechanisms or constraints govern variations in phase modularity (at the macroscopic level accessible to scalp EEG) across a broad range of frequencies. Our goal was to understand the mechanisms that controlled spontaneous fluctuations in phase modularity. Our strategy was to compare the spatial distributions of phase relations between maximally and minimally modular states (defined as top and bottom 5% in $q$). Our analyses revealed a surprisingly simple mechanism.

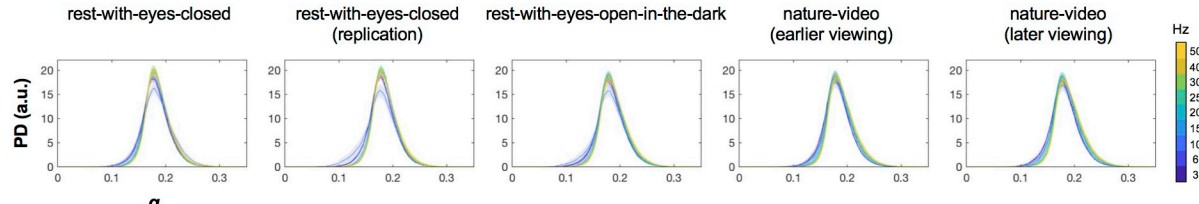

**Fig 1. Temporal distributions of $q$ (maximized modularity in global phase relations) for representative frequencies (3–50 Hz) for the five behavioral conditions.** The y-axis is probability density (a.u.). Frequencies from 3 Hz to 50 Hz are coded with cooler to warmer colors. The shadings around the lines represent the 95% confidence intervals. Note that the $q$ distributions are similar for all representative frequencies and across the five behavioral conditions.

Three spatial networks accounted for 55% to 76% (depending on frequency and condition) of inter-site phase relations in high-$q$ (high-modularity) and low-$q$ (low-modularity) states. The first two networks were characterized by their stability, always maintaining low-$|\Delta\theta|$ ($<\pi/4$) phase relations—the stable low-$|\Delta\theta|$ network—or always maintaining high-$|\Delta\theta|$ ($>3\pi/4$) phase relations—the stable high-$|\Delta\theta|$ network—irrespective of variations in $q$. These may reflect foundational networks due to their stability.

The most interesting was the third "shifting-$|\Delta\theta|$" network that coherently shifted phase relations from low-$|\Delta\theta|$ in low-$q$ states to high-$|\Delta\theta|$ in high-$q$ states, primarily accounting for fluctuations in phase modularity. Whereas the spatial structures of the two stable-$|\Delta\theta|$ networks depended on frequency and behavioral condition, the spatial structure of the shifting-$|\Delta\theta|$ network was largely invariant across frequencies and behavioral conditions, characterized by long-range phase relations between anterior and posterior regions. Thus, the shifting-$|\Delta\theta|$ anterior-posterior network may play a major role in controlling phase modularity by synchronizing (in low-modularity states) or phase-lagging (in high-modularity states) anterior and posterior processes through multiple frequency bands. Inter-site power correlations in the beta and gamma bands were elevated in high-$q$ (high-modularity) states, suggesting that phase modularity impacts neural interactions. These results may motivate further investigations into the functional roles of phase modularity and the shifting-$|\Delta\theta|$ anterior-posterior network that controls it.

## Materials and methods

### Participants

Fifty-two Northwestern University students (35 women, 1 non-binary; mean age of 20.8 years, ranging from 18 to 29 years, standard deviation of 2.5 years) gave informed written consent to participate for monetary compensation ($10/hr). All were right-handed, had normal hearing and normal or corrected-to-normal vision, and had no history of concussion. They were tested individually in a dimly lit or darkened room in the period between 5/11/2017 and 1/27/2020. The data obtained from none of the recruited participants were excluded from analyses. The study protocol was approved by the Northwestern University Institutional Review Board. Participants p1-p7 and p12-p28 ($N = 24$) participated in a rest-with-eyes-closed condition in which EEG was recorded for ~5 min while participants rested with their eyes closed and freely engaged in spontaneous thoughts. Participants p8-p28 ($N = 21$) subsequently participated in a silent-nature-video condition in which EEG was recorded for ~5 min while they viewed a silent nature video. To evaluate test-retest reliability, the silent-nature-video condition was run twice (20–30 min apart), labeled as earlier viewing and later viewing in the analyses. A generic nature video was presented on a 13-inch, 2017 MacBook Pro 2880(H)-by-1800(V)-pixel-resolution LCD monitor with normal brightness and contrast settings, placed 100 cm away from participants, subtending approximately 16˚(H)-by-10˚(V) of visual angle. Participants p29-p52 ($N = 24$) participated in the replication of the rest-with-eyes-closed condition and subsequently participated in a rest-with-eyes-open-in-the-dark condition which was the same as the former except that the room was darkened and participants kept their eyes open while blinking naturally. Subsets or the whole of these data were previously analyzed for different purposes [22–25].

### EEG recording and pre-processing

While participants rested with their eyes closed, rested with their eyes open in the dark, or viewed a silent nature video for approximately 5 min, EEG was recorded from 64 scalp electrodes (although we used a 64-electrode montage, we excluded signals from noise-prone

electrodes, *Fpz*, *Iz*, *T9*, and *T10*, from analyses) at a sampling rate of 512 Hz using a BioSemi ActiveTwo system (see www.biosemi.com for details). Electrooculographic (EOG) activity was monitored using four face electrodes, one placed lateral to each eye and one placed beneath each eye. Two additional electrodes were placed on the left and right mastoids. The EEG data were preprocessed using EEGLAB and ERPLAB toolboxes for MATLAB [26, 27]. The data were re-referenced offline to the average of the two mastoid electrodes, bandpass-filtered at 0.01 Hz-80 Hz, and notch-filtered at 60 Hz (to remove power-line noise that affected the EEG signals from some participants). For the EEG signals recorded while participants rested with the eyes open in the dark or while they viewed a silent nature video, an Independent Component Analysis (ICA) was conducted using EEGLABs' *runica* function [28, 29]. Blink-related components were visually identified (apparent based on characteristic topography) and removed (no more than two components were removed per participant).

## Estimating dura sources by surface-Laplacian transforming EEG signals

To examine inter-regional phase relations of neural activity, it is necessary to apply a source estimation method to scalp EEG potentials to reduce volume-conduction effects that would generate spurious inter-sensor phase relations at 0 and $\pi$ across distances substantially greater than 5 cm [30, 31]. EEG source reconstruction methods constrained by structural MRI and fMRI localizers obtained from each person can achieve superior source reconstruction with models customized for each person [32]. This approach was unavailable to us as we had neither structural MRI nor fMRI data for our participants. Among the non-customized source-imaging methods, we chose the surface-Laplacian transform that (theoretically) estimates the spatial distribution of macroscopic current sources/sinks at the dura surface. The surface-Laplacian transform has been shown to produce similar sources to those inferred by deconvolving surface potentials using a generic model of thicknesses and impedances of scalp and skull [31]. Commonly used source-imaging methods such as sLORETA and Beamforming have been shown to approximate simulated sources and/or to extract neural correlates of behaviors to a similar degree as the surface-Laplacian transform [30, 33]. Further, there is no evidence (to our knowledge) to suggest that these source-imaging methods provide greater spatial resolution than the surface-Laplacian transform. Thus, our preference was to use the latter because it is the most general source-imaging method that relies the least on model-specific assumptions and free parameters [13, 30, 31, 34, 35]. Further, simulations using realistic volume-conduction models have shown that the surface-Laplacian transform can improve phase-relation estimates over analyses of raw EEG potentials [36].

The surface-Laplacian transform is expected to reduce volume-conduction effects to within 1–3 cm [5, 30] which approximately corresponded to the average spacing of sensors in our 64-channel montage. We verified this estimate by showing that, after the application of the surface-Laplacian transform, between-site phase locking precipitously dropped as a function of distance over the range of neighboring sites at rates typically more than an order of magnitude steeper than the rates at which phase locking gradually declined at longer distances (see the Results section).

For our implementation of the surface-Laplacian transform, we used Perrin and colleagues' algorithm [37–39] with a "smoothness" value, $\lambda = 10^{-5}$ (recommended for 64 channels [5]). We refer to the surface-Laplacian transformed EEG signals that represent the macroscopic current sources/sinks on the dura surface under the 60 scalp sites (with the 4 noise-prone sites removed from analyses; see above) simply as EEG signals. These EEG-recording and pre-processing procedures were identical to those used in our prior studies [22, 23].

## EEG analysis

We used the temporal derivative of EEG as in our prior studies that examined all [22, 23] or a subset [25] of the current EEG data for different purposes. While the rationale for taking the temporal derivative of EEG is detailed in [23], it offers the following advantages. First, EEG temporal derivatives may highlight oscillatory dynamics by reducing the non-oscillatory $1/f^{\beta}$ spectral background when β~1, which was generally the case for our EEG data on the timescale of several seconds [23]. Second, EEG temporal derivatives may be considered a "deeper" measure of neural activity than EEG in the sense that scalp-recorded potentials are caused by the underlying neural currents and taking EEG temporal derivative macroscopically estimates those currents (as currents in RC circuits are proportional to the temporal derivative of the corresponding potentials). Third, EEG temporal derivatives are drift free. Prior studies used EEG temporal derivatives for similar reasons [e.g., 40–42], additionally providing some evidence suggesting that EEG temporal derivatives offer neural features that are more effective than EEG for brain-computer interface [42].

To investigate how spectral phase (the phase angle of a sinusoidal component) fluctuated over time, we used a Morlet wavelet-convolution method suitable for time-frequency decomposition of signals containing multiple oscillatory sources of different frequencies (see [5] for a review of different methods for time-frequency decomposition). Each Morlet wavelet is a Gaussian-windowed sinusoidal templet characterized by its frequency as well as its temporal and spectral widths that limit its temporal and spectral resolution. We transformed each EEG waveform (i.e., its temporal derivative) into a time series of wavelet-convolved complex signals (containing spectral amplitude and phase information) using a set of wavelets with 200 frequencies, $f_w$'s, ranging between 3 Hz and 60 Hz. The $f_w$'s were logarithmically spaced because neural temporal-frequency tunings tend to be approximately logarithmically scaled [43, 44]. The accompanying $n$ factor (roughly the number of cycles per wavelet, defined as, $n = 2\pi f \cdot SD$, where $SD$ is the wavelet standard deviation) was also logarithmically spaced between 3 and 16, yielding temporal resolutions ranging from $SD$ = 159 ms (at 3 Hz) to $SD$ = 42 ms (at 60 Hz) and spectral resolutions ranging from $FWHM$ (full width at half maximum) = 2.36 Hz (at 3 Hz) to $FWHM$ = 8.83 Hz (at 60 Hz). These values struck a good balance for the time-frequency trade-off and are typically reported in the literature [5]. While our prior studies focused on spectral power (squared modulus of the complex signals) [22, 23, 25], the current study focused on spectral phase. We examined phase relations at representative frequencies: 3, 6, 10, 15, 20, 25, 30, 40, and 50 Hz. In some cases, we scrutinized frequency dependencies at 1 Hz resolution up to 20 Hz because some features of phase relations were pronounced in the alpha range.

Temporal averages of pairwise inter-site phase relations were computed as complex phase differences, $\left\langle \Delta\theta_{complex} \right\rangle = \left\langle \frac{W_1(t,f)^* W_2(t,f)}{|W_1(t,f)||W_2(t,f)|} \right\rangle$, where $W_1(t,f)$ and $W_2(t,f)$ represent complex signals at site 1 and site 2 obtained by convolving EEG signals with the wavelet of frequency $f$ at time $t$, the asterisk indicates complex conjugation, | | indicates taking the modulus, and $\langle\ \rangle$ indicates temporal averaging. Once time averaged complex phase differences $\langle\Delta\theta_{complex}\rangle$ were computed, they were converted to real phase differences, $\Delta\theta = \text{atan}\left(\frac{Im\left[\langle\Delta\theta_{complex}\rangle\right]}{Re\left[\langle\Delta\theta_{complex}\rangle\right]}\right)$, where $Im$ and $Re$ are the imaginary and real parts, respectively. Phase clustering values (PCVs), indicative of the temporal stability of phase differences, were obtained by taking the modulus of $\langle\Delta\theta_{complex}\rangle$.

## Results and discussion

### High-*q* (high-modularity) states characterized by increased high-|Δθ| relations

As a first step to investigating the networks that drive the phase modularity between high-*q* (high-modularity) and low-*q* (low-modularity) states (see Introduction), we compared inter-site phase differences |Δθ|'s between the periods yielding the top and bottom 5% in *q* (time-averaged within high-*q* and low-*q* periods).

To illustrate the relationship between phase modularity and |Δθ| distributions, we present a representative example of inter-site phase-similarity matrices and |Δθ| distributions in high-*q* and low-*q* states for one participant in the rest-with-eyes-closed condition for 10 Hz. As expected, in high-*q* states, distinct synchronized (low-|Δθ|) clusters were segregated from one another by large phase lags, exhibiting a high degree of modularity (distinct yellow regions segregated by dark-blue regions in Fig 2, upper left panel). In low-*q* states, inter-site synchronizations (low-|Δθ| relations) were increased, but they were not distinctly clustered (Fig 2, lower left panel). These modularity differences were evident in |Δθ| distributions as the relative prevalence of low-|Δθ| and high-|Δθ| relations. In high-*q* states, consistent with the ample dark-blue (high-|Δθ|) regions segregating the yellow clusters, the distribution of |Δθ| was characterized by elevated high-|Δθ| (|Δθ|>3π/4) relations (Fig 2, upper right panel). In low-*q* states, consistent with the increased inter-site synchronizations, the distribution of |Δθ| was characterized by elevated low-|Δθ| (|Δθ|<π/4) relations (Fig 2, lower right panel). The elevated high-|Δθ| relations in high-*q* states and the elevated low-|Δθ| relations in low-*q* states were remarkably consistent across frequencies, behavioral conditions, and participants (Fig 3).

What networks of sites increased high-|Δθ| relations in high-*q* states and increased low-|Δθ| relations in low-*q* states?

### Clusters of site pairs with (1) stable low-|Δθ| phase relations, (2) stable high-|Δθ| phase relations, and (3) shifting-|Δθ| phase relations that shifted from low-|Δθ| in low-*q* (low-modularity) states to high-|Δθ| in high-*q* (high-modularity) states

Before proceeding with the phase-relation analyses, we implemented measures to reduce potential spurious contributions. First, highly distance-dependent phase-coherence at close inter-site distances likely reflects volume-conduction effects instead of neural interactions. For surface-Laplacian transformed EEG data (estimating dura sources), such spurious inter-site synchronization substantially attenuates over the typical distance between adjacent sites for a 64-channel montage [5]. To verify this for our current data set, we examined the scatter plots of pairwise inter-site PCVs (phase clustering values—the moduli of temporal averages of complex |Δθ|'s; see the Methods section) as a function of 3D inter-site distance scaled such that the distances between most neighboring sites were less than $d = 0.5$ ($M = 0.42$, $SD = 0.085$) and the maximum inter-site distance was $d = 2$ (e.g., between two circumferential sites on opposite sides). As expected, PCVs rapidly attenuated within $d = $ minimum to $d = 0.5$ for all representative frequencies for all behavioral conditions. The initial rapid attenuation (which was typically more than an order of magnitude steeper than the subsequent gradual attenuation) was followed by the bona fide reductions in inter-site phase coherence at larger distances (Fig 4). We thus excluded all inter-site pairs with distances less than $d = 0.5$ from the subsequent analyses. We also excluded site pairs with below-median PCVs (per frequency per condition per participant) to focus on phase relations that were relatively stable.

### One participant in the rest-with-eyes-closed condition for 10 Hz

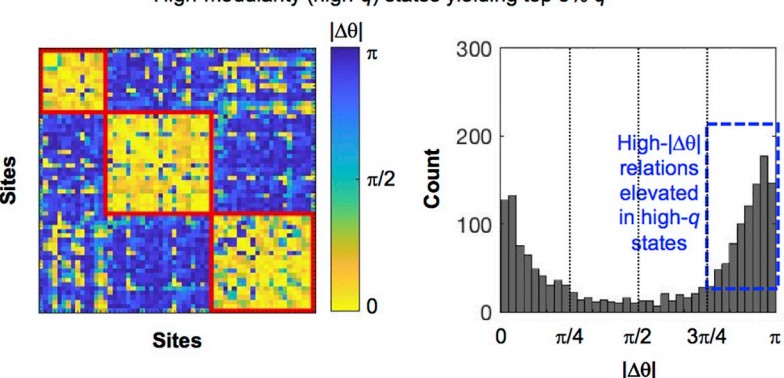

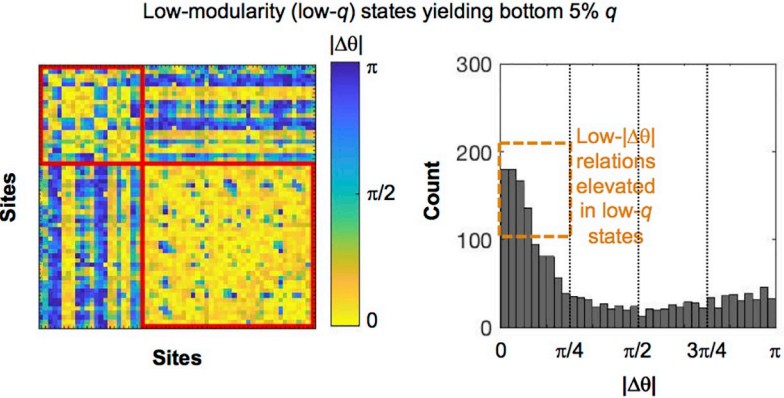

**Fig 2.** Inter-site phase-similarity $|\Delta\theta|$ matrices (left panels) and $|\Delta\theta|$ distributions (right panels) in high-modularity (high-$q$) (upper panels) and low-modularity (low-$q$) (lower panels) states. The index $q$ (maximized modularity) computed per time point per frequency indicates the degree to which within-cluster $|\Delta\theta|$ was lower than between-cluster $|\Delta\theta|$ (see main text). High- and low-modularity states were defined as those yielding $q$ values in the top and bottom 5%, respectively. Inter-site $|\Delta\theta|$ values were separately time-averaged for high-$q$ and low-$q$ states. Here, the 10 Hz data from one participant in the rest-with-eyes-closed condition are shown as an example. In the phase-modularity matrices (left panels), $|\Delta\theta|$ values are color-coded such that yellow tones indicate synchronization (low $|\Delta\theta|$), green tones indicate medium phase lags ($|\Delta\theta|$ around $\pi/2$), and blue tones indicate large phase lags (high $|\Delta\theta|$). In addition, the sites are ordered such that those belonging to the same yellow (i.e., low-$|\Delta\theta|$ or synchronized) clusters are grouped. Note that high-modularity (high-$q$) states (top row) were characterized by elevated high-$|\Delta\theta|$ ($>3\pi/4$) relations (see the large dark-blue regions segregating the distinct yellow clusters in the upper left panel as well as the elevated high-$|\Delta\theta|$ peak in the upper-right panel), whereas low-modularity (low-$q$) states (bottom row) were characterized by elevated low-$|\Delta\theta|$ ($<\pi/4$) relations (see the increased yellow regions in the lower left panel as well as the elevated low-$|\Delta\theta|$ peak in the lower-right panel).

To investigate how phase relations changed between high-$q$ and low-$q$ states, we plotted each site pair as a point with its average $|\Delta\theta|$ in high-$q$ states on the y-axis and its average $|\Delta\theta|$ in low-$q$ states on the x-axis (Fig 5). All site pairs (with above-median PCVs per frequency per condition per participant) from all participants are plotted with their PCVs color-coded (warmer colors representing larger PCVs). Each row-pair in Fig 5 corresponds to a specific behavioral condition with the upper and lower panels color-coding PCVs in high-$q$ and low-$q$ states, respectively; the points in the upper and lower panels would have been identical if we plotted all site pairs regardless of their PCVs. However, the points are slightly different between the upper and lower panels because we excluded site pairs with below-median PCVs;

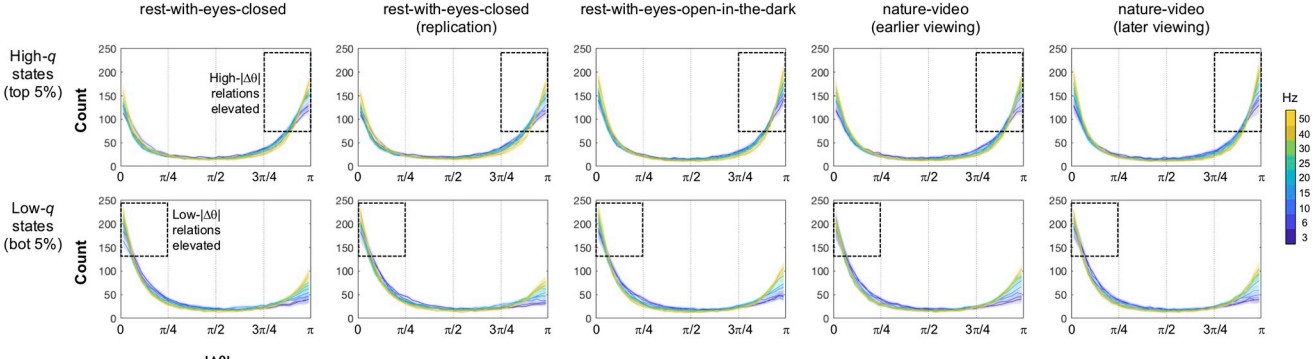

**Fig 3.** Distributions of pairwise inter-site phase differences |Δθ| in high-*q* (top 5%) and low-*q* (bottom 5%) states for representative frequencies (color coded) for the five behavioral conditions (columns). Equivalent to the right panels in Fig 2, but the histograms (generated with 40 angular bins) have been averaged across participants, and are shown for representative frequencies (3–50 Hz, coded with cooler to warmer colors) and for the five behavioral conditions (columns). The shadings around the lines represent the 95% confidence intervals. Note that high-|Δθ| (>3π/4) relations were consistently elevated in high-*q* (high-modularity) states (upper row), whereas low-|Δθ| (<π/4) relations were consistently elevated in low-*q* (low-modularity) states (lower row).

some site pairs dipped below median PCV only in high-*q* states whereas some site pairs did so only in low-*q* states.

Points on the diagonal indicate site pairs that yielded the same average |Δθ| in both high-*q* and low-*q* states. Points above (or below) the diagonal indicate site pairs that yielded higher (or lower) average |Δθ| in high-*q* relative to low-*q* states. For all behavioral conditions (row pairs) and representative frequencies (columns), the points were densely clustered in three specific corners (highlighted with dashed squares in Fig 5). The lower-left and upper-right clusters along the diagonal indicate site pairs that maintained low-|Δθ| (<π/4) and high-|Δθ| (>3π/4) relations, respectively, in both low-*q* and high-*q* states. Consistent with the above observation that high-*q* (high-modularity) states were characterized by increased high-|Δθ| relations (Figs 2 and 3), the third upper-left (off-diagonal) cluster indicates site pairs that shifted their phase relations from low-|Δθ| (<π/4) in low-*q* (low-modularity) states to high-|Δθ| (>3π/4) in high-*q* (high-modularity) states. These three clusters (labeled in the zoomed-in panel at the top of Fig 5) accounted for 55% to 76% of all site pairs depending on frequency and behavioral condition (the percentage of site pairs in each cluster is indicated in Fig 5).

It may be the case that the lower-left clusters of site pairs that maintained low-|Δθ| relations in both low-*q* and high-*q* states always maintained low-|Δθ| relations, forming a *stable* low-|Δθ| network. Similarly, the upper-right clusters of site pairs that maintained high-|Δθ| relations in both low-*q* and high-*q* states may have always maintained high-|Δθ| relations, forming a *stable* high-|Δθ| network. For the upper-left clusters of site pairs that shifted their phase relations from low-|Δθ| in low-*q* states to high-|Δθ| in high-*q* states, it is unclear whether the shifts were bimodal—switching between low-|Δθ| and high-|Δθ| relations—or continuous—shifting gradually between low-|Δθ| and high-|Δθ| relations as a function of *q*. To address this question, we examined the distribution of |Δθ| for each of the three cluster types as a function of *q*. As we defined low-*q* and high-*q* states as the bottom and top 5% in *q*, we examined the distributions of |Δθ| in 5% *q* increments (0–5%, 5–10%, 10–15%,..., up to 95–100%, with the first and last *q* ranges corresponding to low-*q* and high-*q* states, respectively).

In Fig 6, the column triplets correspond to the five behavioral conditions and the rows correspond to representative frequencies. For each column triplet (for a specific condition), the left column shows the |Δθ| distributions for the lower-left clusters of site pairs with low-|Δθ|

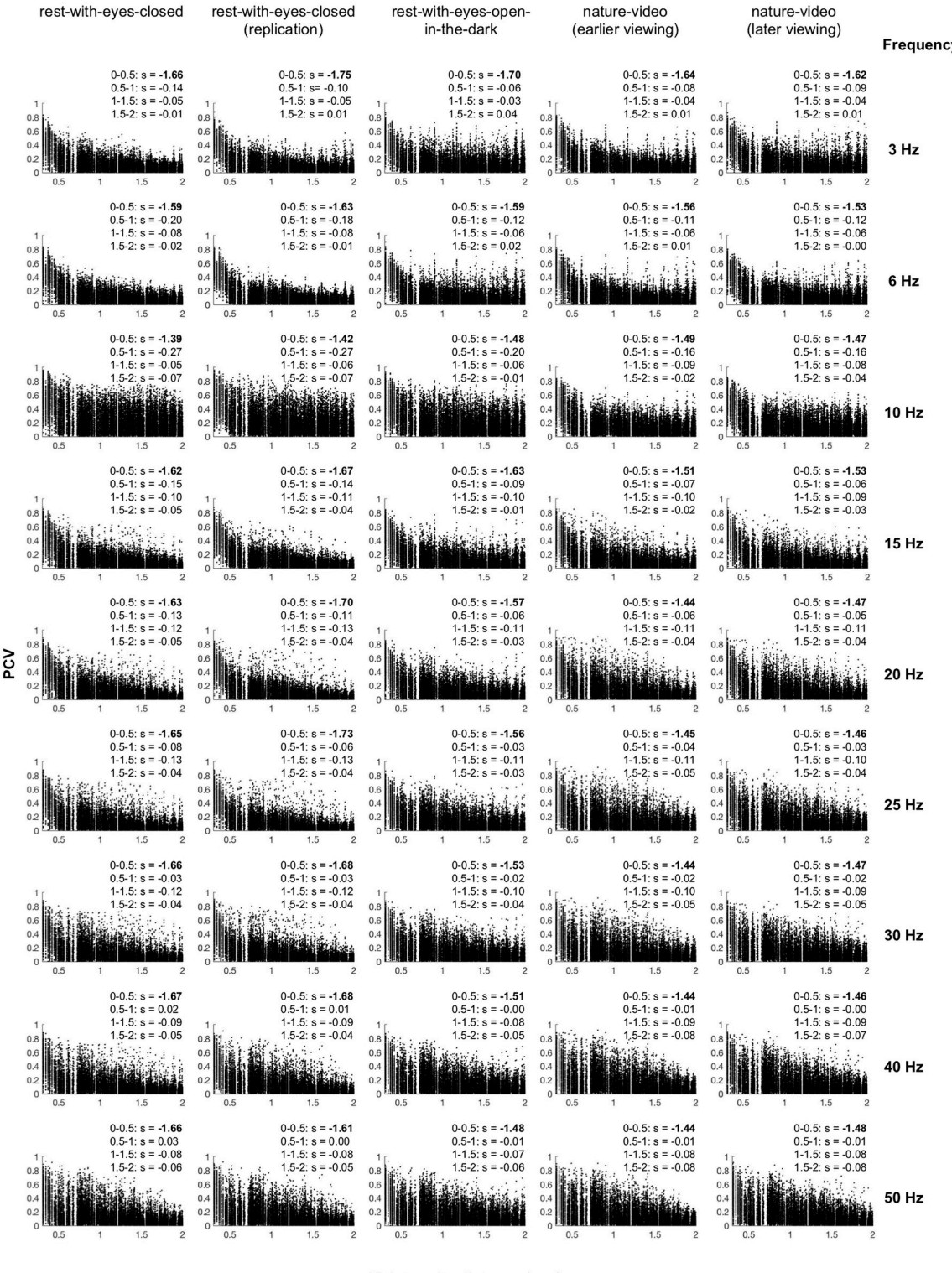

**Fig 4. Scatter plots showing the relationships between PCV (phase clustering value) and 3D inter-site distance for representative frequencies (rows) for the five behavioral conditions (columns).** Each point represents the PCV (computed across all time points, y-axis) for a pair of sites separated by a specific 3D distance (x-axis). Points are shown for all site pairs from all participants for the corresponding frequency (row) and behavioral condition (column). Although the distance is in an arbitrary unit, the distances between the majority of neighboring sites were less than $d = 0.5$ ($M = 0.42$, $SD = 0.085$) and $d = 2$ corresponded to the maximum distance

(between two circumferential sites on opposite sides). In each panel, linear regression slopes are indicated for four equal intervals, $d = 0$–$0.5$, $d = 0.5$–$1$, $d = 1$–$1.5$, and $d = 1.5$–$2$. Note that PCV rapidly attenuated by the distance of $d = 0.5$ as the regression slopes were typically more than an order of magnitude steeper in the first interval than in any other intervals, suggesting that volume-conduction effects were largely confined within neighboring sites as previously reported for surface-Laplacian transformed EEG (see main text).

relations (Fig 5), the middle column shows the |Δθ| distributions for the upper-right clusters with high-|Δθ| relations, and the right column shows the |Δθ| distributions for the upper-left clusters that shifted from low-|Δθ| relations in low-*q* states to high-|Δθ| relations in high-*q* states, with the *q* ranges color-coded (cooler colors for lower-*q* ranges and warmer colors for higher-*q* ranges). The site pairs in the lower-left clusters (Fig 5) always maintained low-|Δθ| relations regardless of *q* for all representative frequencies and behavioral conditions (left columns in Fig 6), forming the stable low-|Δθ| network. Similarly, the site pairs in the upper-right clusters always maintained high-|Δθ| relations regardless of *q* (middle columns in Fig 6), forming the stable high-|Δθ| network. The site pairs in the upper-left clusters bimodally shifted from low-|Δθ| relations in low-*q* ranges to high-|Δθ| relations in high-*q* ranges (right columns in Fig 6) rather than gradually changing |Δθ| as a function of *q*, thus forming the shifting-|Δθ| network. The shifting-|Δθ| network spent more time in high-|Δθ| relations than in low-|Δθ| relations as the distributions leaned toward high-|Δθ| for mid-range *q* (green curves in the right columns in Fig 6).

The stable low-|Δθ| network and the stable high-|Δθ| network may contribute general stability, whereas the shifting-|Δθ| network may control fluctuations in phase modularity between low-*q* and high-*q* states. In the subsequent analyses, we determined whether the three networks exhibited consistent spatial structures across participants, frequencies, and behavioral conditions.

## Spatial structures of the stable low-|Δθ|, stable high-|Δθ|, and shifting-|Δθ| networks

To determine the spatial structures of the stable low-|Δθ|, stable high-|Δθ|, and shifting-|Δθ| networks, we examined the spatial distributions of "connectivity degrees" (per frequency per condition per participant) for each of the three clusters of site pairs (Fig 5). The connectivity degree for a site *k*, $n_k$, was defined as the number of site pairs from a specific cluster (e.g., stable low-|Δθ| cluster) that included site *k*, which is equivalent to the usual definition [14]. We defined connectivity degrees relative to the values expected by chance (assuming the relevant site pairs were randomly distributed), that is, $n_k = n_{k(actual)} − n_{k(expected\ by\ chance)}$. Thus, a large positive connectivity degree would indicate a strong involvement of the corresponding site in the network defined by the relevant cluster (e.g., stable low-|Δθ| cluster) potentially playing the role of a network hub, whereas a large negative connectivity degree would indicate the site not being a part of the network.

Topographic heatmaps of connectivity degrees for the low-|Δθ|, high-|Δθ|, and shifting-|Δθ| phase networks are shown in Fig 7 as *t*-values against zero, with warmer colors representing above-chance values and cooler colors representing below-chance values, with the warmest and coolest colors indicating the Bonferroni-corrected two-tailed statistical significance at α = 0.05 (corresponding to *t*-values of ±4, correcting for 60 comparisons at 60 sites). The same heatmaps with connectivity degrees plotted as proportions of deviations from the chance levels (rather than *t*-values) are shown in S1 Fig.

In Fig 7, the column triplets correspond to the five behavioral conditions and the row pairs correspond to representative frequencies. For each column triplet (for a specific condition),

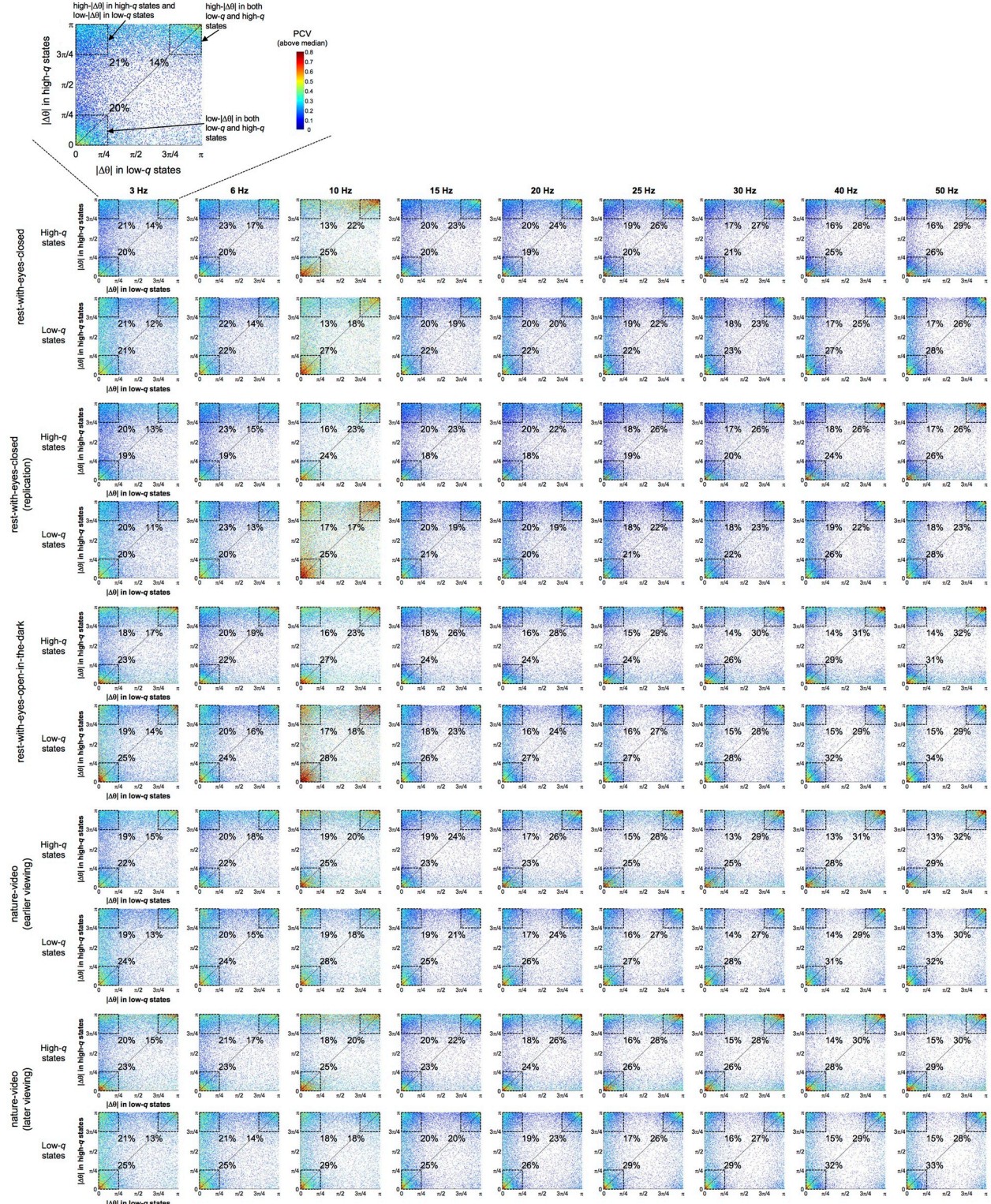

**Fig 5. Scatter plots showing the relationship between inter-site |Δθ| in high-*q* (y-axis) and low-*q* (x-axis) states for representative frequencies (columns) and the five behavioral conditions (row pairs).** A zoomed-in version is shown for the upper-left panel. Each point represents a site pair that yielded above-median PCV (per frequency per participant per condition) with the y-value indicating the average |Δθ| in high-*q* states and the x-value indicating the average |Δθ| in low-*q* states. Points on the diagonal indicate site pairs that yielded the same average |Δθ| in both high-*q* and low-*q* states. Points above (or below) the diagonal indicate site pairs that yielded higher (or lower) average |Δθ| in high-*q* relative to low-*q* states. The

PCV for each site pair is color coded (warmer colors for larger PCV). For each behavioral condition, the upper panel shows PCVs in high-$q$ states and the lower panel shows PCVs in low-$q$ states. The patterns are slightly different in these panels because of the missing points that dipped below median PCV in high-$q$ or low-$q$ states. Note that the points are densely clustered in the three corner regions highlighted with dashed squares, indicating that large proportions of site pairs either maintained low-$|\Delta\theta|$ ($<\pi/4$) (lower-left squares) or high-$|\Delta\theta|$ ($>3\pi/4$) (upper-right squares) relations in both low-$q$ and high-$q$ states, or shifted their phase relations from low-$|\Delta\theta|$ ($<\pi/4$) in low-$q$ states to high-$|\Delta\theta|$ ($>3\pi/4$) in high-$q$ states (upper-left squares). These dense clusters accounted for 55%-76% of all site pairs (depending on frequency and behavioral condition).

the left column shows the heatmaps of connectivity degrees for the stable low-$|\Delta\theta|$ network (corresponding to the cluster highlighted with the lower-left squares in Fig 5), the middle column shows the heatmaps of connectivity degrees for the stable high-$|\Delta\theta|$ network (corresponding to the clusters highlighted with the upper-right squares in Fig 5), and the right column shows the heatmaps of connectivity degrees for the shifting-$|\Delta\theta|$ network (corresponding to the clusters highlighted with the upper-left squares in Fig 5). For each row pair (for a specific frequency), the upper and lower plots show the heatmaps of connectivity degrees in high-$q$ and low-$q$ states, respectively; these plots are slightly different from each other because of the missing site pairs that dipped below median PCV in high-$q$ or low-$q$ states (as in Fig 5). Note that the shifting-$|\Delta\theta|$ network changed phase relations from low-$|\Delta\theta|$ in low-$q$ states (upper rows) to high-$|\Delta\theta|$ in high-$q$ states (lower rows).

As mentioned above, the network hubs (with significantly elevated connectivity degrees) are indicated by dark-red regions in Fig 7. For the stable low-$|\Delta\theta|$ (left columns) and stable high-$|\Delta\theta|$ (middle columns) networks, the hubs were not strongly clustered below 10 Hz (relatively large regions in green to orange colors at 3 and 6 Hz), but uniquely clustered at 10 Hz, and progressively more consistently clustered above 10 Hz. Specifically, at 3 and 6 Hz, the stable low-$|\Delta\theta|$ and stable high-$|\Delta\theta|$ networks were characterized only by the exclusions of posterior (left columns at 3 and 6 Hz) and anterior (middle columns at 3 and 6 Hz) sites, respectively. At 10 Hz (highlighted with a horizontal rectangle), the stable low-$|\Delta\theta|$ network (left columns) included anterior hubs in all conditions, whereas the stable high-$|\Delta\theta|$ network (middle columns) included the central-anterior and bilateral-posterior hubs only in the conditions with minimal visual input (Fig 7A–7C to some degree). At 15–50 Hz, the stable low-$|\Delta\theta|$ network progressively more consistently included the central-mid hub and circumferentially distributed hubs (primarily covering anterior and lateral regions) (left columns at 15 Hz and above), whereas the stable high-$|\Delta\theta|$ network progressively more consistently included the semi-circularly (primarily in anterior and lateral regions) to circularly distributed hubs surrounding the central-mid region (middle columns at 15 Hz and above). Thus, the networks that stably maintained low-$|\Delta\theta|$ or high-$|\Delta\theta|$ phase relations regardless of $q$ exhibited distinct macroscopic spatial organizations that systematically depended on frequency.

Notably, the spatial network that shifted phase relations from low-$|\Delta\theta|$ in low-$q$ states to high-$|\Delta\theta|$ in high-$q$ states was invariant across frequencies and conditions. This shifting-$|\Delta\theta|$ network was consistently characterized by the bilateral anterior and posterior hubs (Fig 7, right columns highlighted with gray rectangles), except at 10 Hz for the minimal-visual-input conditions where the posterior hubs were absent (Fig 7A–7C, right columns).

## Connectivity characteristics (distance, orientation, and region) of the stable low-$|\Delta\theta|$, stable high-$|\Delta\theta|$, and shifting-$|\Delta\theta|$ networks

While the topographic heatmaps of connectivity degree (Fig 7) show network hubs, they do not show the patterns of connectivity. Here, we use terms such as "connectivity" and "connections" merely to describe the spatial networks of site pairs that consistently maintained low-$|\Delta\theta|$ or high-$|\Delta\theta|$ phase relations with above-median PCVs in low-$q$ and/or high-$q$ states

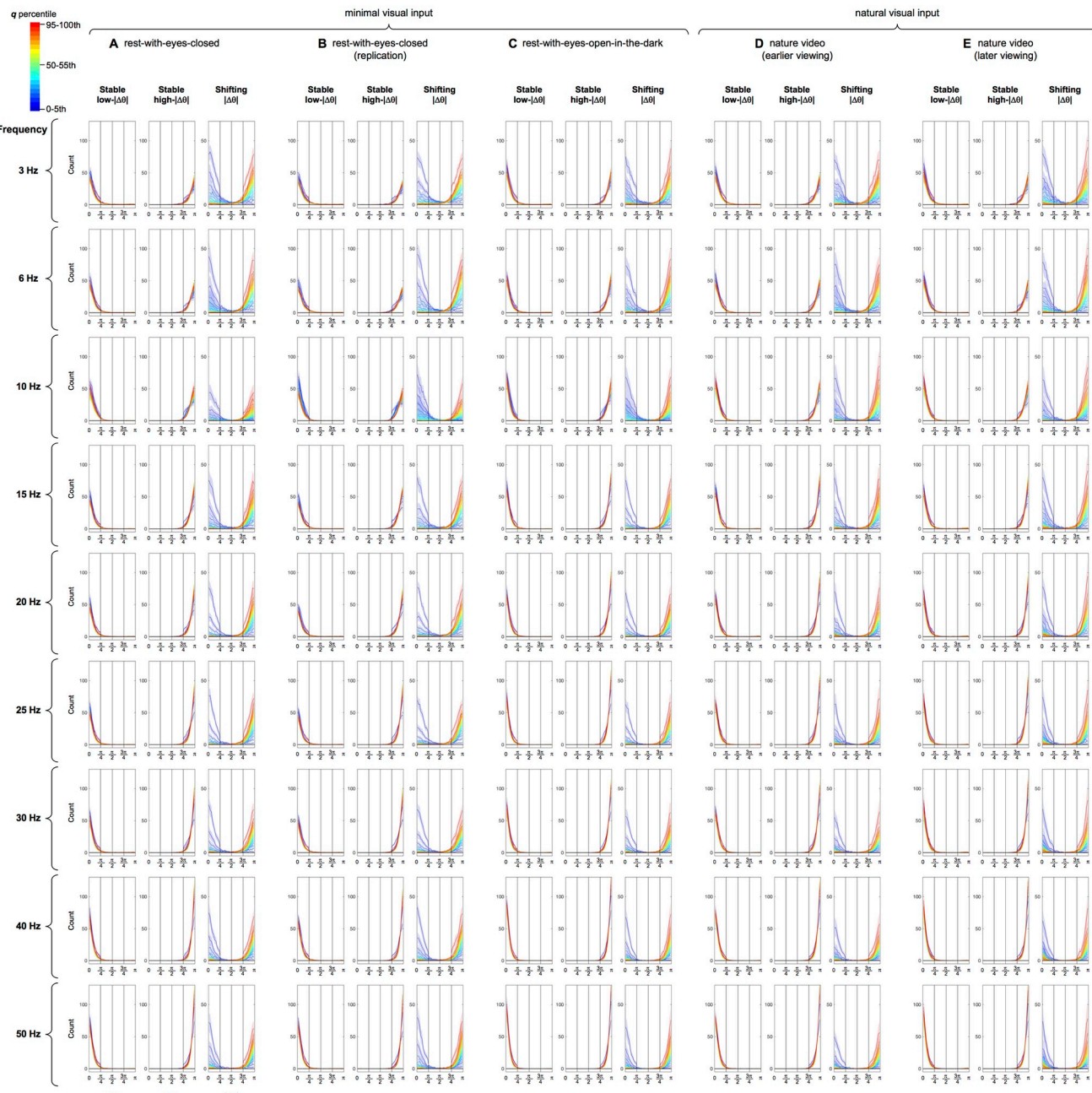

**Fig 6. Distributions of |Δθ| for the stable low-|Δθ| (left columns), stable high-|Δθ| (middle columns), and shifting-|Δθ| (right columns) networks for different $q$ ranges in 5-percentile increments (color coded), for representative frequencies (rows) and the five behavioral conditions (column triplets labeled A-E).** The histograms include site pairs with above-median PCVs and were made with 40 equal angular bins. **Left column for each condition.** Distributions of |Δθ| for the site pairs that remained low-|Δθ| in both low-$q$ and high-$q$ states (corresponding to the site-pair clusters highlighted with the lower-left squares in Fig 5). These sites formed the stable low-|Δθ| network as their phase relations always remained low-|Δθ| regardless of $q$ values. **Middle column for each condition.** Distributions of |Δθ| for the site pairs that remained high-|Δθ| in both low-$q$ and high-$q$ states (corresponding to the site-pair clusters highlighted with the upper-right squares in Fig 5). These sites formed the stable high-|Δθ| network as their phase relations always remained high-|Δθ| regardless of $q$ values. **Right column for each condition.** Distributions of |Δθ| for the site pairs that shifted their phase relations from low-|Δθ| in low-$q$ states to high-|Δθ| in high-$q$ states (corresponding to the site-pair clusters highlighted with the upper-left squares in Fig 5). Note that these sites bimodally shifted their phase relations between low-|Δθ| and high-|Δθ| values as a function of $q$. The shadings around the lines represent the 95% confidence intervals.

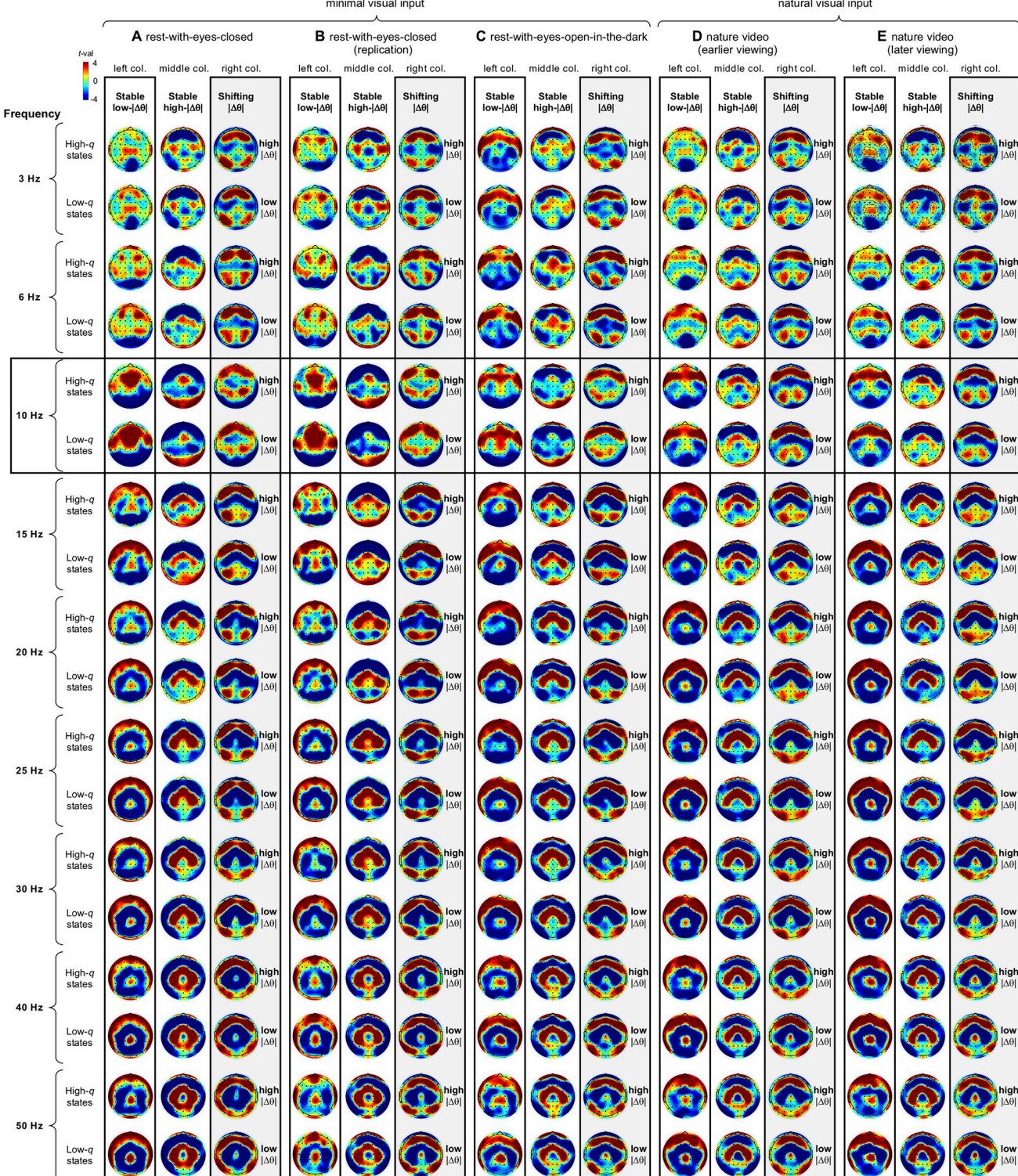

**Fig 7. Spatial distributions of connectivity degree for the stable low-|Δθ| (left columns) and stable high-|Δθ| (middle columns) networks, and for the shifting-|Δθ| network that shifted its phase relations from low-|Δθ| in low-*q* (low-modularity) states to high-|Δθ| in high-*q* (high-modularity) states (right, highlighted columns), for representative frequencies (row pairs) and the five behavioral conditions (column triplets labeled A-E). Left column for each condition**. Spatial distributions of connectivity degree for the stable low-|Δθ| network that always maintained low-|Δθ| relations regardless of *q*. For each site (per frequency per condition per participant) its connectivity degree was computed as the number of sites with which it maintained stable low-|Δθ| relations minus the number expected by chance (assuming the site pairs constituting the stable low-|Δθ| network were randomly distributed); a large

positive value indicates a "hub" status (substantially above-chance connectivity degree with the stable low-|Δθ| network) whereas a large negative value indicates an "out-of-network" status (substantially below-chance connectivity degree with the stable low-|Δθ| network). The topographic heatmaps show *t*-values (warmer colors for above-chance values and cooler colors for below-chance values) with the upper and lower limits corresponding to the Bonferroni-corrected two-tailed statistical significance at $\alpha$ = 0.05 (see the color bar at the upper-left corner). The same heatmaps with connectivity degrees indicated as proportions of deviations from the chance levels (rather than *t*-values) are shown in S1 Fig. As in Fig 5, the heatmaps are slightly different for high-*q* (upper rows) and low-*q* (lower rows) states because of the missing site pairs that dipped below median PCV in high-*q* or low-*q* states. **Middle column for each condition**. Spatial distributions of connectivity degree for the stable high-|Δθ| network that always maintained high-|Δθ| relations regardless of *q*. **Right column for each condition**. Spatial distributions of connectivity degree for the shifting-|Δθ| network that shifted their phase relations from low-|Δθ| in low-*q* (low-modularity) states to high-|Δθ| in high-*q* (high-modularity) states. Note that the shifting-|Δθ| network (high-|Δθ| in high-*q* states and low-|Δθ| in low-*q* states) exhibited a consistent spatial structure, characterized by bilateral anterior and posterior hubs regardless of frequency or condition (except at 10 Hz in the minimal-visual-input conditions, A-C).

without implying functional connectivity. To qualitatively visualize connectivity characteristics, we depict representative connections with lines connecting the relevant site pairs, where connections were defined as the stable low-|Δθ|, stable high-|Δθ|, or shifting-|Δθ| phase relations within the respective network (with above-median PCV) that were shared by more than half of the participants (Fig 8). To quantify the connectivity characteristics, we analyzed the prevalence of connections of different distances, different 2D orientations (relative to the anterior-posterior axis), and different types—those within anterior regions, those within posterior regions, those spanning anterior and posterior regions, and those bridging the two cerebral hemispheres, as a function of frequency.

We considered three inter-site distance ranges. The distance of *d* = 0.5 (longer than the majority of distances between neighboring sites) over which the sharply distance-dependent volume-conduction effects substantially attenuated (Fig 4) served as the reasonable minimum distance. We then divided the full range of 3D inter-site distances (maximum at *d* = 2) into three equal ranges: short (*d* = 0.5–1), medium (*d* = 1–1.5), and long (*d* = 1.5–2).

For each distance range, we computed the numbers of within-anterior connections (including those involving the middle row of sites), within-posterior connections (including those involving the middle row of sites), anterior-posterior connections (excluding those involving the middle row of sites), and cross-hemisphere connections (excluding those involving the middle column of sites), per network (stable low-|Δθ|, stable high-|Δθ|, or shifting-|Δθ|) per condition per participant as a function of frequency. The possible numbers of connections differed for the four types in the three distance ranges (due to the configuration of our 64-channel EEG recording cap). We thus corrected the numbers of connection types per distance range (as a function of frequency, *f*) by computing, $n_{(type,dist).corrected}(f) = (n_{(type,dist)}(f) - n_{(type,dist).expected}(f)) + N_{dist}(f)/4$, where $n_{(type,dist)}(f)$ is the actual number of relevant (stable low-|Δθ|, stable high-|Δθ|, or shifting-|Δθ|) connections of a specific type (e.g., within-anterior) for a given distance range (e.g., *d* = 1–1.5), $n_{(type,dist).expected}(f)$ is the expected value of $n_{(type,dist)}(f)$ obtained if the total number $N_{dist}(f)$ of the relevant connections in the given distance range were randomly distributed, with $N_{dist}(f)/4$ (with 4 being the number of connection types) ensuring that the sum of $n_{(type,dist).corrected}(f)$ across all four connection types added up to $N_{dist}(f)$. Random connectivity would yield $n_{(type,dist).corrected}(f) = N_{dist}(f)$ (plus some random fluctuations) for all connection types for a given distance range.

For each distance range, we also computed the average 2D orientation of connections. To assess a potential longitudinal versus horizontal bias in connections, we computed the absolute azimuth angle that the line connecting each relevant site pair made with the anterior-posterior axis, ranging from 0˚ (parallel to the anterior-posterior axis; i.e., vertical/longitudinal) to 90˚ (perpendicular to the anterior-posterior axis; i.e., horizontal). For each distance range, we first computed an orientation histogram with six evenly spaced orientation bins (per network per condition per participant as a function of frequency) and corrected the number per orientation

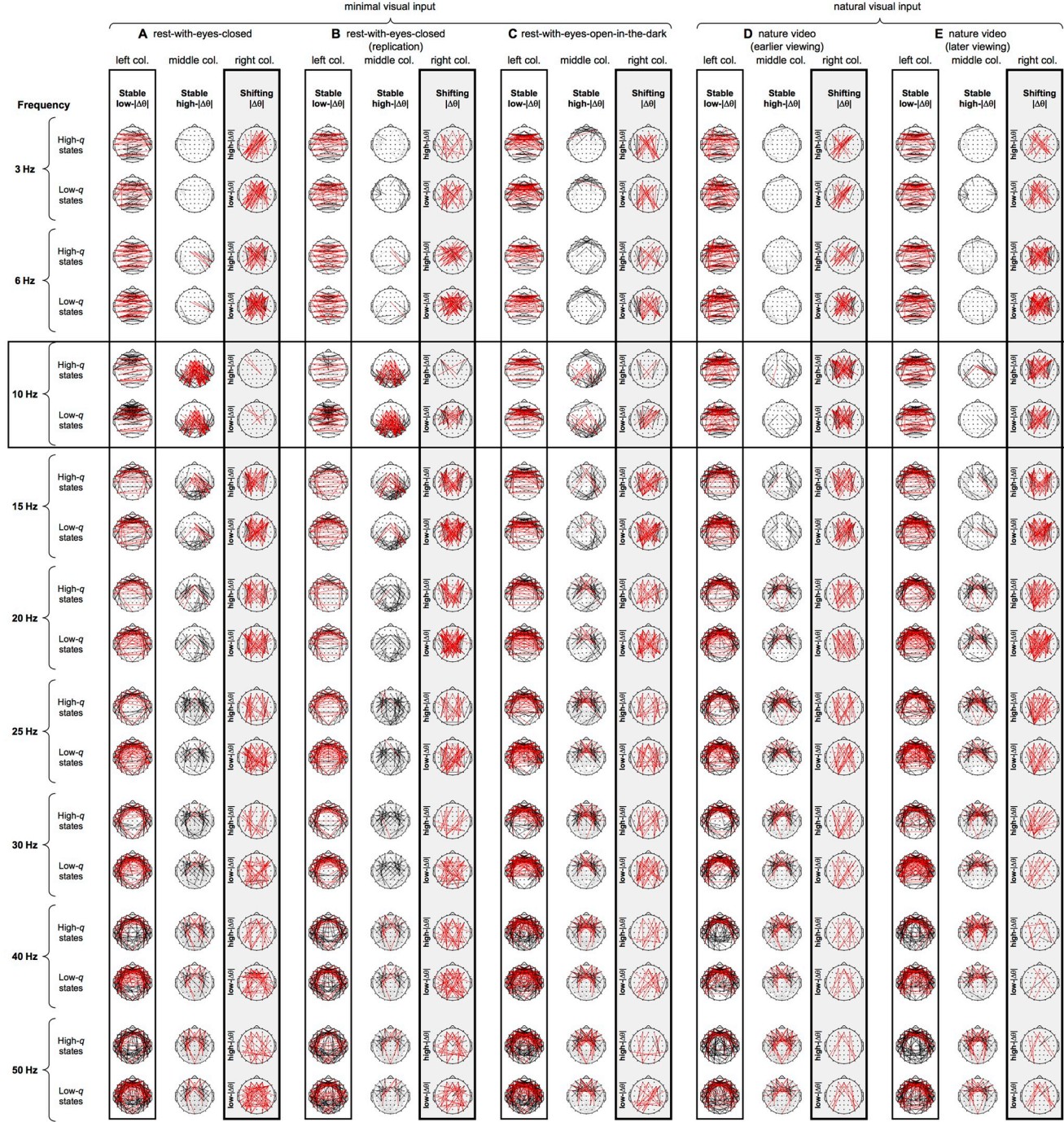

**Fig 8. Representative connections in the stable low-|Δθ| (left columns), stable high-|Δθ| (middle columns), and shifting-|Δθ| (right, highlighted columns) networks, for representative frequencies (row pairs) and the five behavioral conditions (column triplets).** The lines represent site pairs with above median PCVs (per frequency per condition per participant) within each network (stable low-|Δθ|, stable high-|Δθ|, or shifting-|Δθ|), which we refer to as "connections," that were shared by at least half of the participants (N>12 for the three rest conditions and N>11 for the two nature video conditions). Short- ($d$ = 0.5–1), medium- ($d$ = 1–1.5), and long- ($d$ = 1.5–2) distance connections are indicated with gray, black, and red lines, respectively. **Left column for each condition**. Representative connections in the stable low-|Δθ| network. As in Figs 5 and 7 the representative connections were slightly different for high-$q$ (upper rows) and low-$q$ (lower rows) states because of the missing site pairs that dipped below median PCV in high-$q$ or low-$q$ states. **Middle column for each condition**. Representative connections in the stable high-|Δθ| network. **Right column for each condition (highlighted)**. Representative connections in the shifting-|Δθ| network, that shifted from low-|Δθ| in low-$q$ states to high-|Δθ| in high-$q$ states. Note the prevalence of anterior-posterior long-distance connections (red lines) in the shifting-|Δθ| network.

bin using the formula introduced above (replacing the four connection types with six orientation bins). We then computed the average orientation from each histogram. Note that the intermediate step of constructing an orientation histogram was necessary for correcting for the differences in the possible numbers of connections of different orientations in different distance ranges. Although average orientations around 45˚ could potentially reflect orientation peaks around 45˚, we verified that those cases instead reflected flat distributions of orientation (not shown). Thus, average orientations of <45˚ indicated biases toward vertical/longitudinal orientation, >45˚ indicated biases toward horizontal orientation, and ~45˚ indicated an even distribution of connection orientations.

The layout of the qualitative connectivity plots (Fig 8) parallels that of the connectivity degree heatmaps (Fig 7). The five behavioral conditions are organized as column triplets with each column (within a triplet) showing the connections in the stable low-|Δθ| (left column), stable high-|Δθ| (middle column), and shifting-|Δθ| (right column) networks. Frequencies are organized as row pairs with the upper and lower rows corresponding to high-$q$ and low-$q$ states; as in Fig 7, the connectivity maps are slightly different for high-$q$ and low-$q$ states because of the missing site pairs that dropped below median PCV (per frequency per participant) in high-$q$ or low-$q$ states.

The layout of the panels showing the prevalence statistics (averages and 95% confidence intervals) of the connectivity features (distance, orientation, and region) is slightly different (Fig 9). In Fig 9A–9C, each column triplet corresponds to a network type (stable low-|Δθ|, stable high-|Δθ|, or shifting-|Δθ|) while the columns within each triplet show the connectivity feature statistics for the short (left columns), medium (middle columns), and long (right columns) distance connections. The behavioral conditions are organized as row pairs (the upper and lower rows still corresponding to high-$q$ and low-$q$ states); as above, the connectivity-feature statistics are slightly different for high-$q$ and low-$q$ states because of the missing site pairs that dipped below median PCV (per frequency per participant) in high-$q$ or low-$q$ states. Within each panel, the average numbers of within-anterior (red), within-posterior (blue), anterior-posterior (purple), and cross-hemisphere (green) connections are plotted as a function of frequency. For the panels showing the average 2D orientations of connections (Fig 9D), each column corresponds to a specific network (the stable low-|Δθ| [left], stable high-|Δθ| [middle], or shifting-|Δθ| [right] network), and connection distance is coded with line thickness: thin lines for short, medium-think lines for medium, and thick lines for long distance connections. Note that, because we corrected the actual counts of the various connection types in each distance range for their possible maximum numbers (given the scalp-site configuration used in the current study), all curves within each panel would overlap and the average 2D orientation would be 45˚ in all cases if the relevant connections were randomly distributed.

## Connectivity characteristics of the stable low-|Δθ| network

For the stable low-|Δθ| network at 10 Hz and below, horizontal medium- and long-distance connections, especially in anterior regions, dominated with only sparse longitudinal medium- and long-distance connections (the black and red lines in Fig 8, left columns, top three row pairs). The quantitative results confirmed the prevalent anteriorly biased horizontal medium- and long-distance connections at lower frequencies, showing elevated anterior and cross-hemisphere connections (the elevated red and green curves in Fig 9A, middle and right columns) with horizontal orientation biases (the peaked medium-thick and thick curves in Fig 9D, left column) at lower frequencies (< ~15 Hz), especially in the alpha range. The quantitative results also confirmed the sparse longitudinal connections at lower frequencies, showing a dip in anterior-posterior medium- and long-distance connections below ~15 Hz (the purple

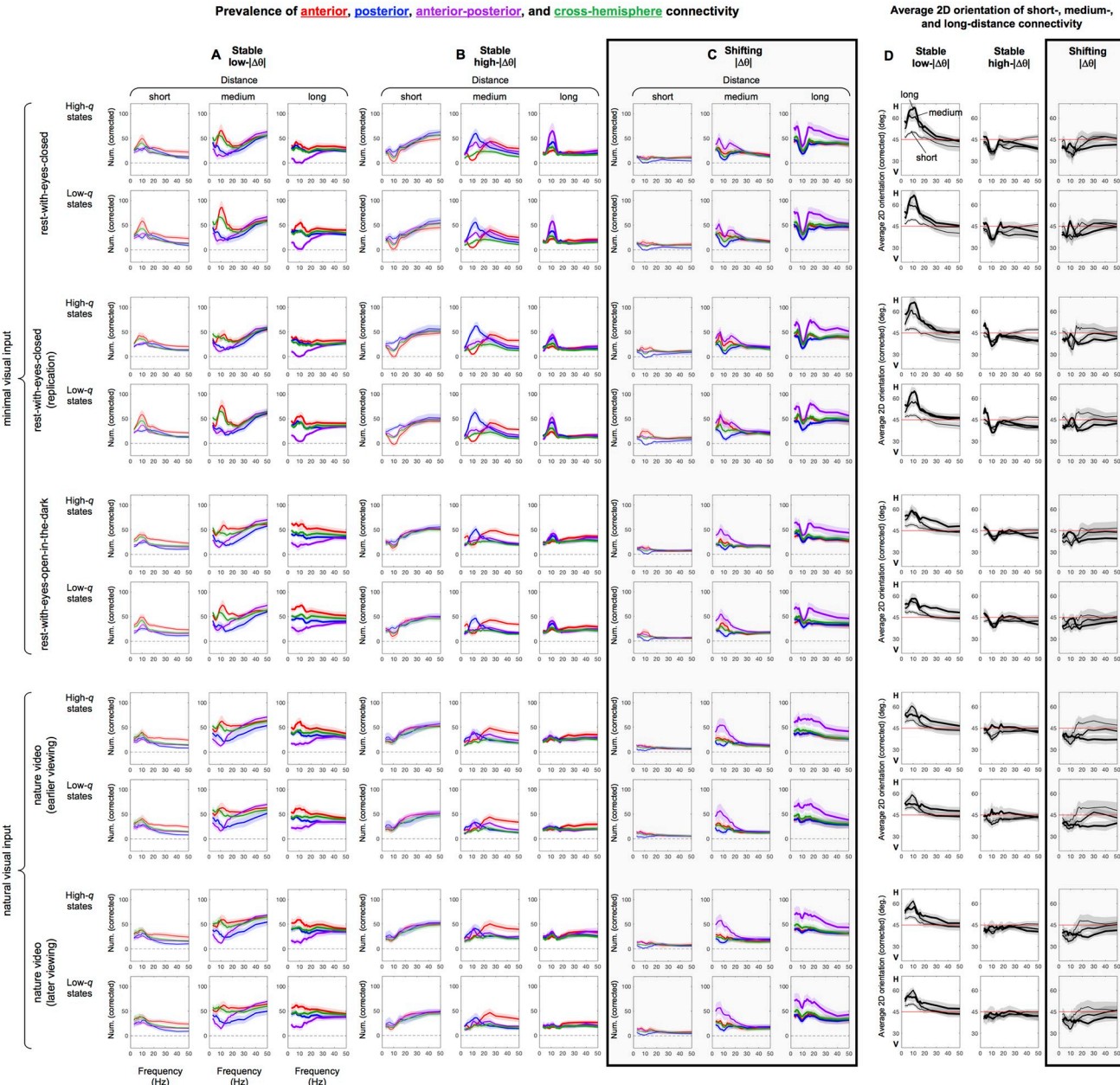

**Fig 9. Connectivity characteristics (distance, orientation, and region) of the stable low-|Δθ|, stable high-|Δθ|, and shifting-|Δθ| networks. A**. The prevalence of the short- ($d$ = 0.5–1), medium- ($d$ = 1–1.5), and long- ($d$ = 1.5–2) distance connections (columns, also coded with line thickness) of the four types (within-anterior [red], within-posterior [blue], anterior-posterior [purple], and cross-hemisphere [green]) as a function of frequency for the stable low-|Δθ| network. The counts have been corrected to account for the differences in the available numbers of connections of different distances and types (see main text). With the correction, stochastic (e.g., phase-scrambled) data would generate overlapping curves within each panel. The row pairs correspond to the five behavioral conditions. As in Figs 5, 7 and 8, the patterns were slightly different for high-$q$ (upper rows) and low-$q$ (lower rows) states because of the missing site pairs that dipped below median PCV in high-$q$ or low-$q$ states. **B**. The prevalence of the short-, medium-, and long-distance connections (columns, also coded with line thickness) of the four types (color coded) as a function of frequency for the stable high-|Δθ| network. **C**. The prevalence of the short-, medium-, and long-distance connections (columns, also coded with line thickness) of the four types (color coded) as a function of frequency for the shifting-|Δθ| network. **D**. Average 2D orientations of short- (thin curves), medium- (medium-thick curves), and long- (thick curves) distance connections (relative to the anterior-posterior axis) with >45˚ indicating a horizontal bias, <45˚ indicating a vertical/longitudinal bias, and 45˚ indicating no orientation bias, for the stable low-|Δθ|, stable high-|Δθ|, and shifting-|Δθ| networks (columns). For all panels, the shadings around the lines represent the 95% confidence intervals. Note that the shifting-|Δθ| network was consistently dominated by anterior-posterior long-distance connections (the purple curves dominant in C, right column) that were longitudinally biased (the thick curves generally <45˚ in D, right column) for all frequencies and conditions, except in the alpha range in the minimal-visual-input conditions (the purple curves dip in C, right column, and the thick curves peak in D, right column, in the alpha range in the top three row pairs).

curves in Fig 9A, middle and right columns). Overall, the elevated horizontal connections (especially in the anterior region) combined with the reduced longitudinal connections suggest that the stable low-|Δθ| network at lower frequencies (< ~15 Hz) is primarily lateral and segregated between anterior and posterior regions.

At higher (> 15 Hz) frequencies, medium- and long-distance connections (black and red lines) across circumferential sites as well as medium-distance connections (black lines) between central-mid and circumferential sites in all orientations are notable, with the central-mid-to-circumferential medium-distance connections (black lines) increasing at higher frequencies (Fig 8, left columns, at 20 Hz and above). These connections are consistent with the central-mid and circumferential hubs (Fig 7, left columns, at 20 Hz and above). The quantitative results generally confirmed these observations as medium-distance connections increased above ~20 Hz (all curves monotonically increased above ~20 Hz, Fig 9A, middle column) as well as connections became increasingly omni-orientational above ~20 Hz (most curves approached 45˚ above ~20 Hz in Fig 9D, left column).

**Connectivity characteristics of the stable high-|Δθ| network.** For the stable high-|Δθ| network at 10 Hz (highlighted with the horizontal rectangle), dense long-distance connections (red lines) between central-anterior and bilateral-posterior regions and medium-distance connections (black lines) within the posterior region are notable in the three minimal-visual-input conditions (Fig 8A–8C, middle columns at 10 Hz), consistent with the central-anterior and bilateral-posterior hubs (Fig 7A–7C, middle columns at 10 Hz). The quantitative results confirmed these observations as anterior-posterior long-distance connections (the purple curves in Fig 9B, right column) and posterior medium-distance connections (the blue curves in Fig 9B, middle column) both peaked in the alpha range in the minimal-visual-input conditions (top three row pairs), with biases toward the longitudinal orientation (the medium-thick and thick curves dipping below 45˚ in the alpha range in Fig 9D, middle column, in the top three row pairs). These connections were reduced in the eyes-open conditions (the reduced blue peaks in Fig 9B, middle column, and the absent purple peaks in Fig 9B, right column, in the bottom two row pairs).

At frequencies 20 Hz and above, dense, somewhat longitudinally biased, medium-distance connections (black lines) were prevalent in lateral-anterior regions (Fig 8, middle columns, at 20 Hz and above), consistent with the upper-semicircular hubs surrounding the central-mid region (Fig 7, middle columns, at 20 Hz and above). The quantitative results confirmed these observations as anterior medium-distance connections increased above ~20 Hz (the red curves elevated above ~20 Hz in Fig 9B, middle column) that were slightly biased toward the longitudinal orientation (the medium-thick curves generally lower than 45˚ above ~20 Hz in Fig 9D, middle column). Also notable is the rapid increase in short-distance connections of all orientations above 20 Hz (the increasing number of gray lines above 20 Hz in Fig 8, middle columns). This was also confirmed by the quantitative results as short-distance connections generally increased above ~20 Hz (all curves rising above ~20 Hz in Fig 9B, left column) and they tended to be omni-orientational (the thin lines generally around 45˚ above ~20 Hz in Fig 9D, middle column).

The quantitative results additionally indicate that the long-distance connections were regionally balanced and frequency independent (all curves generally overlapping and flat above ~20 Hz in Fig 9B, right column), and either orientationally unbiased or slightly biased toward longitudinal (the thick curves generally around or below 45˚ above ~20 Hz in Fig 9D, middle column). One may observe that the representative long-distance connections shown in the connectivity plots at 20 Hz and above (the red lines in Fig 8, middle columns, at 20 Hz and above) appear to be biased toward anterior regions rather than being regionally balanced as indicated by the quantitative results. Because we defined representative connections in the

connectivity plots (Fig 8) as those shared by more than half of the participants, the seeming discrepancy suggests that the locations of the long-distance connections for the high-|Δθ| network were more variable across participants in posterior regions than in anterior regions above ~20 Hz.

**Connectivity characteristics of the shifting-|Δθ| network.** The connectivity characteristics were especially consistent for the shifting-|Δθ| network. Long-distance connections (red lines) bilaterally bridging anterior and posterior sites dominated at all frequencies (Fig 8, right columns), consistent with the robust bilateral anterior and posterior hubs (Fig 7, right columns), except at 10 Hz in the minimal-visual-input conditions. In the latter case, the posterior hubs were absent (Fig 7A–7C, right columns, at 10 Hz) and the longitudinal long-distant connections were either minimal or did not reach posterior regions (Fig 8A–8C, right columns, at 10 Hz). The quantitative results confirmed these observations. Anterior-posterior long-distance connections dominated (the elevated purple curves in Fig 9C, right column) with consistent biases toward the longitudinal orientation (the thick curves consistently below 45˚ in Fig 9D, right column), except in the alpha range in the minimal-visual-input conditions (the purple curves dipping in Fig 9C, right column, combined with the thick curves peaking toward 45˚ in Fig 9D, right column, in the alpha range in the top three row pairs). Overall, the shifting-|Δθ| network was consistently characterized by bilateral anterior-posterior long-distance connections for all frequencies (except in the alpha range) and behavioral conditions, suggesting that this network may play a general role in regulating long-distance anterior-posterior communications.

## Discussion

Much research has focused on identifying networks formed by on-demand phase realignments as neural correlates of specific behaviors [e.g., 8, 10–12, 15, 16]. We focused on a complementary goal of characterizing the rules governing the general dynamics of global phase relations using scalp-recorded EEG. We reasoned that global phase relations may spontaneously fluctuate between more and less organized states. We further reasoned that an organized state may be characterized by modularity, that is, by distinct clusters of synchronized neural populations interacting with one another at large phase lags, exhibiting small within-cluster inter-site |Δθ|'s with large between-cluster |Δθ|'s. We thus tracked fluctuations in phase modularity using the index $q$ (quantifying the degree to which |Δθ|'s are low within each cluster and high between the clusters, varying from 0 to 1) calculated with a standard community-detection algorithm [14, 21].

Phase modularity, $q$, computed in this way exhibited temporal distributions that were invariant across a broad range of frequencies (3–50 Hz) and behavioral conditions (Fig 1). When we compared inter-site phase relations between high-$q$ (top 5th percentile) and low-$q$ (bottom 5th percentile) states, we discovered surprisingly simple characteristics. Substantial proportions of site pairs maintained either low-|Δθ| ($<\pi/4$) relations (19%-33% of site pairs, depending on frequency and behavioral condition) or high-|Δθ| ($>3\pi/4$) relations (11%-32% of site pairs) in both low-$q$ and high-$q$ states (Fig 5). These site pairs formed the stable low-|Δθ| network and the stable high-|Δθ| network, respectively, as they always maintained their low-|Δθ| or high-|Δθ| relations regardless of $q$ (Fig 6). Another substantial proportion of site pairs (13%-23%) formed the shifting-|Δθ| network as they coherently shifted their phase relations from low-|Δθ| ($<\pi/4$) in low-$q$ (low-modularity) states to high-|Δθ| ($>3\pi/4$) in high-$q$ (high-modularity) states in a bimodal manner (Figs 5 and 6).

The stable low-|Δθ| network comprised site pairs that always maintained synchronization (in the sense that low |Δθ|$<\pi/4$ phase relations may be considered synchronization). Although

volume-conduction effects are a concern when evaluating low-|Δθ| relations, we are reasonably confident that the stable low-|Δθ| network reflects inter-site synchronization over and above any volume-conduction effects as we only considered site pairs separated by $d > 0.5$ while the strongly distance-dependent component of PCV (likely reflective of volume-conduction effects) substantially attenuated by $d = 0.5$ (Fig 4; also consistent with the expected spatial resolution of surface-Laplacian transformed EEG using the standard 64-channel configuration; e.g., [5]). Further, the distinguishing features of the stable low-|Δθ| network primarily reflected medium- and long-distance ($d \gg 0.5$) connections, making contributions from volume-conduction effects even less likely (Figs 7–9). As mentioned in the results section, we refer to the consistent phase relations within the stable low-|Δθ|, stable high-|Δθ|, and shifting-|Δθ| networks (that were above-median) as connectivity or connections with no deeper connotations.

At lower frequencies ($< \sim 15$ Hz), the stable low-|Δθ| (synchronization) network was characterized by elevated, horizontally-biased within-anterior connections combined with reduced longitudinal connections, suggestive of an anterior-posterior segregation. At higher frequencies ($> \sim 20$ Hz) this network was characterized by omni-orientational connections emanating from the circumferential and central-mid hubs (Figs 7–9).

The stable high-|Δθ| network comprised site pairs that always maintained large |Δθ|$> 3\pi/4$ phase relations. Some of the high-|Δθ| phase relations, especially at shorter distances, may reflect current-source-and-sink relations corresponding to the same underlying neural processes (e.g., [30]).

Two types of high-|Δθ| medium-to-long distance connections were notable in the alpha range in the minimal-visual-input (especially the eyes-closed) conditions. First, the prevalent connections that bridged central-anterior and bilateral posterior regions likely reflect anterior-posterior interactions with high-|Δθ| (these long-distance connections are unlikely to reflect current-source-and-sink pairs). Second, the prevalent medium-distance connections in posterior regions may reflect hierarchical interactions with high-|Δθ| (e.g., between lower and higher visual areas) as the connections were consistently longitudinally biased; we are unaware of any *a priori* reasons to expect that the current-source-and-sink pairs in posterior regions would be consistently longitudinally biased. Thus, when visual input is minimal (in the eyes-closed and eyes-open-in-the-dark conditions), the posterior alpha processes that are hierarchically interacting with high-|Δθ| may be controlled or integrated by central-anterior alpha processes through long-distance interactions with high-|Δθ|.

At higher frequencies ($> \sim 20$ Hz), the stable high-|Δθ| network was characterized by the steady increase in regionally and orientationally balanced short-distance connections at higher frequencies, medium-distance connections emanating from the anteriorly biased hubs surrounding the central-mid region, and regionally and orientationally balanced long-distance connections. Thus, larger numbers of current-source-sink assemblies may be detected with scalp-recorded EEG for higher-frequency oscillations (which the prevalent short-distance connections may reflect) and that those assemblies located in the anteriorly biased regions surrounding the central-mid region may interact with neighboring and distant assemblies with high-|Δθ|.

Overall, we note that low-|Δθ| connectivity generally tended to be horizontally biased especially at lower frequencies (Fig 9D, left column), whereas high-|Δθ| connectivity tended to be somewhat longitudinally biased or omni-orientational (Fig 9D, middle column). If we broadly consider horizontally versus longitudinally biased interactions as contributing to lateral versus hierarchical neural coordination, the results may suggest that the stable low-|Δθ| network tends to contribute to lateral neural coordination especially at lower frequencies whereas the stable high-|Δθ| network tends to contribute to hierarchical neural coordination.

The discovery of the shifting-|Δθ| network is the most consequential contribution of this study. The basic spatial structure of the shifting-|Δθ| network, namely, the longitudinal, long-distance connectivity bilaterally bridging the anterior and posterior hubs, was invariant across frequencies and behavioral conditions (except in the alpha range in the minimal-visual-input conditions). As high-$q$ (high-modularity) states were primarily characterized by substantially increased high-|Δθ| relations (Figs 2 and 3), the shifting-|Δθ| network drove global phase relations between low-$q$ and high-$q$ states by bimodally (Fig 6) shifting its phase relations from low-|Δθ| in low-$q$ states to high-|Δθ| in high-$q$ states. Specifically, the shifting-|Δθ| network decreased phase modularity in low-$q$ states by synchronizing anterior and posterior processes whereas it increased phase modularity in high-$q$ states by adding large phase lags between anterior and posterior processes. Further, the shifting-|Δθ| network switched anterior-posterior phase relations relatively independently through different frequency bands as we observed little inter-frequency temporal correlations in $q$ (Fig 10).

What functional roles might the anterior-posterior shifting-|Δθ| network play? Understanding this requires an investigation of the potential functional roles of high-$q$ (high-modularity) and low-$q$ (low-modularity) states. While future research needs to address this question, examining power correlations may provide some clues. If neural communications are more facilitated in high-$q$ or low-$q$ states, inter-site temporal correlations in spectral power may be consistently higher in high-$q$ or low-$q$ states. We focused on long-distance ($d$ = 1.5–2) interactions as those would be especially unlikely to be affected by volume-conduction effects. Indeed, inter-site power correlations were generally higher in high-$q$ states than in low-$q$ states for all three networks and in all five behavioral conditions (Fig 11, top row). Nevertheless, the average powers were also higher in high-$q$ states than in low-$q$ states (Fig 11, middle row), raising the possibility that the higher inter-site power correlations in high-$q$ states may simply reflect greater signal-to-noise ratios. This seems unlikely, however. First, the inter-site power correlations in low-$q$ states consistently differed for the three networks, highest for the stable low-|Δθ| network, intermediate for the stable high-|Δθ| network, and lowest for the shifting-|Δθ| network in each condition (the blue curves in Fig 11, top row). However, the corresponding average powers did not follow the same pattern (the blue curves in Fig 11, middle row). Second, if the higher inter-site temporal power correlations primarily reflected greater signal-to-noise ratios in power, the site pairs with higher average powers should yield higher temporal power correlations especially in low-$q$ states where average powers were lower (less possibility of a ceiling effect). This inter-site-pair correlation between temporal power correlation and average power hovered around zero in low-$q$ states (the blue curves in Fig 11, bottom row).

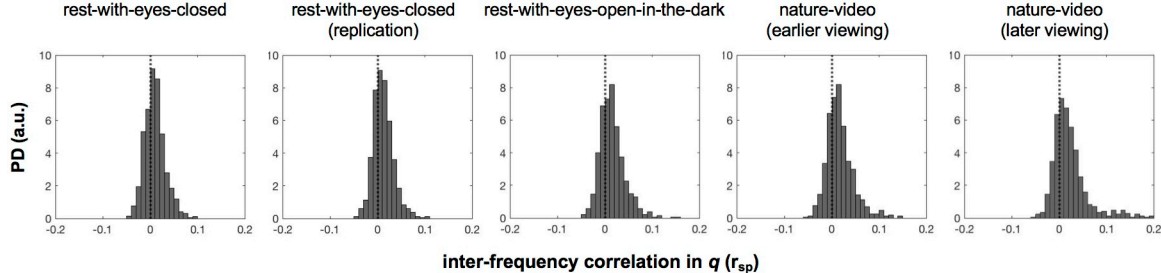

**Fig 10. Distributions of inter-frequency temporal correlations in $q$ for the five behavioral conditions.** Pairwise temporal correlations in $q$ (Spearman's r, or $r_{sp}$) were computed between representative frequencies (3, 6, 10, 15, 20, 25, 30, 40, and 50 Hz) per participant per behavioral condition. Each panel shows the distribution of all $r_{sp}$ values for a specific condition. The y-axis is probability density (a.u.). Note that all correlations were low (approximately centered around zero and mostly less than 0.1).

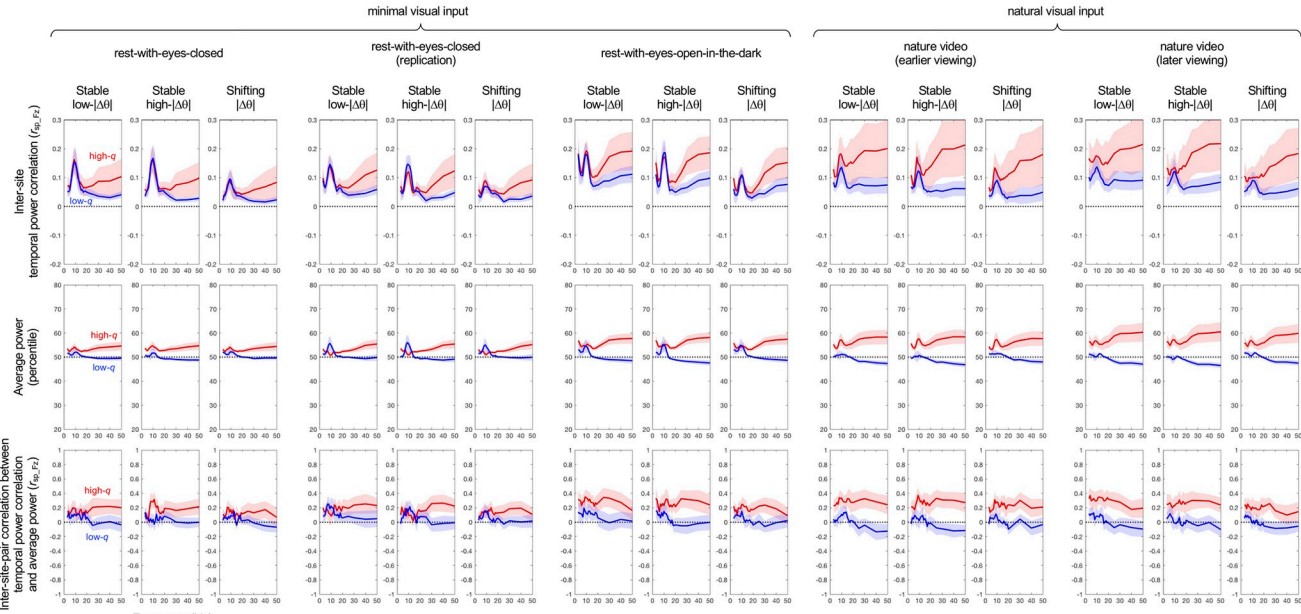

**Fig 11. Inter-site temporal power correlations (top row), average powers (middle row), and inter-site-pair correlations between inter-site power correlation and average power (bottom row) in high-*q* (red) and low-*q* (blue) states for long-distance connections (*d* = 1.5–2) in the stable low-|Δθ| (left columns), stable high-|Δθ| (middle columns), and shifting-|Δθ| (right columns) networks as a function of frequency for the five behavioral conditions (column triplets). Top row**. Inter-site temporal power correlations (in Fisher-*z* transformed Spearman's *r*) in high-*q* (red) and low-*q* (blue) states, averaged separately for the three networks, plotted as a function of frequency. Higher temporal power correlations may suggest stronger long-distance neural interactions. **Middle row**. Powers averaged within each of the three networks in high-*q* and low-*q* states as a function of frequency. The unit is percentile rank (computed per frequency per site per condition per participant); although average powers tended to be higher in high-*q* states than in low-*q* states in all cases, the power elevations were modest as the maximum average power was only about 60th percentile. **Bottom row**. Inter-site-pair correlations between inter-site power correlation and average power in high-*q* and low-*q* states in the three networks as a function of frequency. The correlations were generally positive in high-*q* states indicating that the inter-site power correlations were higher for site pairs with higher average spectral powers in high-*q* states. The correlations hovered around zero in low-*q* states indicating that the inter-site power correlations were relatively independent of average power in low-*q* states. Only long-distance (*d* = 1.5–2) site pairs were included in this analysis as their interactions would unlikely reflect volume-conduction effects. We also did not impose the above-median PCV criterion for this analysis so that the site pairs included in the three networks were identical between low-*q* and high-*q* states. The shadings around the lines represent the 95% confidence intervals.

Given that the inter-site power correlations and average powers are unlikely to be trivially coupled, we make the following observations. The inter-site power correlations were generally higher in high-*q* states than in low-*q* states for frequencies above about 20 Hz (the red vs. blue curves in Fig 11, top row), suggesting that long-distance neural communications in the beta and gamma ranges were facilitated in high-*q* states. The correlation data also confirmed that long-distance power interactions were generally elevated in the alpha range [25] regardless of *q* (all curves peaking in the alpha range in Fig 11, top row). While the average powers remained in the middle range at the 45th to 60th percentile in both low-*q* and high-*q* states, they were moderately but consistently higher in high-*q* states than in low-*q* states (the red vs. blue curves in Fig 11, middle row), suggesting that regional neural synchronizations were moderately elevated in high-*q* states. Finally, while inter-site power correlation was uncorrelated with average power across site pairs in low-*q* states (the blue curves hovering around zero in Fig 11, bottom row), they were moderately correlated in high-*q* states (the red curves generally above zero in Fig 11, bottom row). Taken together, this analysis suggests that high-*q* (high-modularity) states are associated with increased long-distance neural communications in the beta and gamma ranges, with the elevated long-distance interactions being moderately yoked to stronger regional synchronizations.

How is high modularity in phase relations, the defining feature of high-$q$ states, related to elevated long-distance communications in the beta and gamma range? Answering this question requires future research since, to our knowledge, no prior studies examined the dynamics of modularity in global phase relations. It would also be fruitful to more generally investigate the potential functional roles of high-$q$ (high-modularity) versus low-$q$ (low-modularity) states. For example, brief sensory stimuli may be presented at randomly chosen times to evaluate their detectability, discriminability, identifiability, etc., as a function of the $q$ values immediately preceding, during, and/or following stimulus presentations. Attention and/or cognitive performance may also be evaluated/diminished in relation to fluctuations in $q$ values.

Our results are limited in the sense that we evaluated phase relations that were consistent over time, that is, we examined phase relations that remained consistent across the periods of high-$q$ or low-$q$ states. There could be event-related high-$q$ and/or low-$q$ states driven by specific task demands, that may not be controlled by the shifting-$|\Delta\theta|$ network. Alternatively, the shifting-$|\Delta\theta|$ network may always contribute to controlling the modularity in global phase relations. Our results are also limited to within-frequency phase relations while cross-frequency phase coherence has been implicated in attentional and cognitive processes [8, 10, 15, 45]. The usual caveats associated with sinusoidal decompositions of electrophysiological signals (e.g., spurious harmonics generated by non-sinusoidal electrophysiological signals [46–48]) also apply here. Thus, the phase-relation patterns we observed reflect sinusoidal approximations to potentially non-sinusoidal oscillatory signals with the possibility that higher-frequency components partially reflect harmonics generated by non-sinusoidal lower-frequency oscillations. Finally, one may note the prevalence of low ($|\Delta\theta|<\pi/4$) and high ($|\Delta\theta|>3\pi/4$) phase relations ($|\Delta\theta|$) in our results. Some of these may reflect volume-conduction effects and current-source-and-sink pairs, respectively, especially at short distances as discussed above. Interestingly, however, a computational model of a whole-brain network of delay-coupled oscillators built with white-matter connectivity inferred from the connectome data, also yielded phase relations clustered around in-phase ($\Delta\theta = 0$) and anti-phase ($\Delta\theta = \pi$) relations [49].

## Conclusion

We sought to uncover general rules governing the dynamics of global phase relations in oscillatory neural activity reflected in EEG by focusing on phase modularity quantified with the standard index $q$ (maximized modularity [14, 21] computed based on inter-site phase similarity, $|\Delta\theta|$). We identified a shifting-$|\Delta\theta|$ network comprising bilateral anterior-posterior long-distance connectivity that controlled phase modularity by consistently shifting its phase relations from low-$|\Delta\theta|$ in low-$q$ (low-modularity) states to high-$|\Delta\theta|$ in high-$q$ (high-modularity) states through a broad range of frequency bands (3–50 Hz, except in the alpha range in the minimum-visual-input condition) whether participants rested with the eyes closed or viewed a silent nature video. Long-distance neural interactions in the beta and gamma ranges were increased in high-$q$ (high modularity) states, suggesting that phase modularity is functionally consequential. Future research may investigate the mechanisms through which phase modularity is associated with elevated neural interactions as well as investigate the potential functional roles of phase modularity by examining perceptual, attentional, and cognitive performances in relation to $q$ values.

## Supporting information

**S1 Fig. Spatial distributions of connectivity degree for the stable low-$|\Delta\theta|$ (left columns) and stable high-$|\Delta\theta|$ (middle columns) networks, and for the shifting-$|\Delta\theta|$ network that shifted its phase relations from low-$|\Delta\theta|$ in low-$q$ (low-modularity) states to high-$|\Delta\theta|$ in**

**high-$q$ (high-modularity) states (right, highlighted columns), for representative frequencies (row pairs) and the five behavioral conditions (column triplets labeled A-E).** This is the same as Fig 7 except that connectivity degrees are indicated as proportions of deviations from the chance levels (rather than $t$-values). For instance, a value of 0.3 (or -0.3) would indicate that the corresponding connectivity degree was 30% more (or less) than expected by chance. See the color bar in the upper left corner.
(TIF)

**S1 File. The MATLAB code we used to run the standard community-detection algorithm [14, 21] to quantify phase modularity (at each time point per frequency per condition per participant) as the index $q$ (maximized modularity; $0 \leq q \leq 1$).**
(M)

## Author Contributions

**Conceptualization:** Satoru Suzuki.

**Data curation:** Melisa Menceloglu.

**Formal analysis:** Melisa Menceloglu, Satoru Suzuki.

**Funding acquisition:** Melisa Menceloglu.

**Investigation:** Melisa Menceloglu, Marcia Grabowecky, Satoru Suzuki.

**Methodology:** Melisa Menceloglu, Satoru Suzuki.

**Project administration:** Marcia Grabowecky, Satoru Suzuki.

**Resources:** Marcia Grabowecky, Satoru Suzuki.

**Software:** Satoru Suzuki.

**Supervision:** Marcia Grabowecky, Satoru Suzuki.

**Validation:** Satoru Suzuki.

**Visualization:** Satoru Suzuki.

**Writing – original draft:** Satoru Suzuki.

**Writing – review & editing:** Melisa Menceloglu, Marcia Grabowecky, Satoru Suzuki.

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
