## [Decision Letter · Decision Letter 0]

25 Apr 2022

PONE-D-22-00971A phase-­shifting anterior-­posterior network organizes global phase relationsPLOS ONE

Dear Dr. Suzuki,

Thank you for submitting your manuscript to PLOS ONE. After careful consideration, we feel that it has merit but does not fully meet PLOS ONE’s publication criteria as it currently stands. Therefore, we invite you to submit a revised version of the manuscript that addresses the points raised during the review process.

Your draft has been evaluated by the two reviewers. Both of them including myself found

the article very interesting and informative for the field.

The comments raised by the two reviewers can be addressed if you wish to revise the

manuscript.

We look forward to receiving your revised manuscript.

Kind regards,

Stavros I. Dimitriadis

Academic Editor

PLOS ONE

Journal Requirements:

National Institutes of Health T32 NS047987 to MM

Additional Editor Comments:

Dear authors

Your draft has been evaluated by the two reviewers. Both of them including myself found

the article very interesting and informative for the field.

The comments raised by the two reviewers can be addressed if you wish to revise the

manuscript.

Please, respond to every comment raised by the reviewers point by point.

If you wish an extension of the deadline for submitting your revision,

you can send an email.

Reviewers' comments:

Reviewer's Responses to Questions

**Comments to the Author**

1. Is the manuscript technically sound, and do the data support the conclusions?

Reviewer #1: Partly

Reviewer #2: Partly

2. Has the statistical analysis been performed appropriately and rigorously? 

Reviewer #1: Yes

Reviewer #2: Yes

3. Have the authors made all data underlying the findings in their manuscript fully available?

Reviewer #1: Yes

Reviewer #2: Yes

4. Is the manuscript presented in an intelligible fashion and written in standard English?

Reviewer #1: Yes

Reviewer #2: Yes

5. Review Comments to the Author

Reviewer #1: The importance of phase-difference distribution (and of the single values associated with specific pairs of regions) is undubt. The authors investigated global phase-differences to understand the general context in a large cohort of EEG subject, and they found very interesting results. The manuscript is also well written. Nevertheless, I found a few (potential) issues in analysis (1-the fact that it is a scalp level, instead of being at source level, 2-the fact that the phase-differences are computed as the differences between separately calculated single phases, instead of directly computing the phase-difference e.g. from cross-spectral quantities, and 3-the fact that an orthogonalization-procedure to avoid artificial instantaneous coupling has not been applied, thus generating distributions with most of the mass concentrated close to 0 and pi). Prior to publish the manuscript, I would thus suggest a major revision.

Introduction: the whole section (except for the first lines) does not seem to be an introduction of a paper but a synthesis of the methods, results and discussion sections together; I suggest the authors to reshape the introduction in order to focus on the reasons the justified this manuscript. Note: the reasons are somehow there already, but the whole paper would benefit from a deepening into the modern literature, e.g. are there studies focusing on lag/phase-difference distributions? Did the authors of the previously published studies investigate all the frequency bands? Are there other limits in those research papers? Instead just a few articles have been cited so far.

Methods: I would suggest to perform the analysis at source level (i.e. after the application of an inverse method able to estimate source activity from channel level data) in order to further reduce possible artifactual confounds and to better and directly characterise the spatial features at cortical level. If the authors do not consider the application of a source level analysis necessary, they might at least discuss why the applied scalp level analysis does not suffer from usual confounds.

Methods: were the phase-differences calculated as the differences between the computed single phases? I suggest to directly compute the phase-difference instead of separately computing the two single phases to be subtracted (this is an issue that led a lot of researchers to compute the phase-difference from the cross-spectrum and not from the two single Fourier-phases).

Methods: by looking at the |deltatheta|-distributions, it seems that the mass of the (magnitude of the) phase differences is basically either close to 0 or close to pi (U-shaped). This feature may reflect the fact that volume conduction is still playing a confounding role. I suggest to regress-out the contribution of one signal from the second signal before calculating the phase-difference; this process is known as “orthogonalization” and it is also applied to phase-based (but not robust to volume conduction) connectivity metrics. If the authors do not believe this could be the case, they may just show (as examples) that using the orthogonalization does not change the phase-difference distribution for a few site-pairs (or explain that it can introduce additional confound, instead of improving the analysis).

Pag 3: within-cluster |delta theta|’s-> within-cluster |delta theta|. Same for between-cluster…

Page 3: This suggests that modularity q is a fundamental parameter characterizing global phase relations in EEG: I am not quite sure about the truthfulness of this statement. For instance, it is also possible that the modularity q is not suitable to highlight potential global differences among frequency bands or conditions (exaggerating: metrics that always follow a standard normal distribution independently on the input variables might also be useless for characterizing two different input-populations) and the fact that it is invariant under frequency/condition does not make it a “fundamental” parameter.

I would like to thank the authors for their effort; I am sure that handling the above reported issues may additionaly improve their manuscript.

Reviewer #2: In the present manuscript, the authors present a quite interesting approach for modeling phase shifting among anterior - posterior sensor based networks using phase modularity for different individual frequencies. The manuscript is well-written however quite long making it difficult to focus and understand the results. This is more than clear from the figures even though they present interesting results. In this respect, the authors need to take care the figure visibility because it is difficult to make clear interpretations based on the low quality of the figures.

One question is how the authors verify that the presented connectivity results can be supported by source analysis based phase modularity since it is not entirely clear how the volume conduction effects are suppressed. What about the inter-subject variability of skull conductivity for which bone is quite thick at the anterior and posterior regions.

6. PLOS authors have the option to publish the peer review history of their article (what does this mean?). If published, this will include your full peer review and any attached files.

Reviewer #1: No

Reviewer #2: No

---

## [Author Response · Author response to Decision Letter 0]

7 Jun 2022

Please see the cover letter for our point-by-point response to the reviewers' helpful comments. Thank you.

---

## [Decision Letter · Decision Letter 1]

27 Sep 2022

PONE-D-22-00971R1

A phase-shifting anterior-posterior network organizes global phase relations

PLOS ONE

Dear Dr. Suzuki,

Thank you for submitting your manuscript to PLOS ONE. After careful consideration, we have decided that your manuscript does not meet our criteria for publication and must therefore be rejected.

Specifically:

Dear authors

I have read carefully your paper, and I regret to inform you that it is not appropriate for publication.

I totally agree with the comments raised by the reviewer.

You have to narrow down your findings into a specific direction, and also to reduce the size of the draft.

Secondly, you must be careful when you use the term 'diminish' the volume conduction effects, forward problem,

the use of BESA etc.

I definitely recommend to resubmit the work after carefully addressing the suggestions.

I am sorry that we cannot be more positive on this occasion, but hope that you appreciate the reasons for this decision.

Kind regards,

Stavros I. Dimitriadis

Academic Editor

PLOS ONE

Additional Editor Comments :

Dear authors

I have read carefully your paper, and I regret to inform you that it is not appropriate for publication.

I totally agree with the comments raised by the reviewer.

You have to narrow down your findings into a specific direction, and also to reduce the size of the draft.

Secondly, you must be careful when you use the term 'diminish' the volume conduction effects, forward problem,

the use of BESA etc.

I definitely recommend to resubmit the work after carefully addressing the suggestions.

Reviewers' comments:

Reviewer's Responses to Questions

**Comments to the Author**

1. If the authors have adequately addressed your comments raised in a previous round of review and you feel that this manuscript is now acceptable for publication, you may indicate that here to bypass the “Comments to the Author” section, enter your conflict of interest statement in the “Confidential to Editor” section, and submit your "Accept" recommendation.

Reviewer #2: (No Response)

2. Is the manuscript technically sound, and do the data support the conclusions?

Reviewer #2: No

3. Has the statistical analysis been performed appropriately and rigorously? 

Reviewer #2: No

4. Have the authors made all data underlying the findings in their manuscript fully available?

Reviewer #2: No

5. Is the manuscript presented in an intelligible fashion and written in standard English?

Reviewer #2: No

6. Review Comments to the Author

Reviewer #2: The results are difficult to interpret. The reader gets lost in the long and difficult-to-read+interpret results. Also, the authors misuse terms like BESA / sLoreta (one is a commercial tool for source analysis while the other is an inverse approach for reconstructing brain activity. None of sLoreta or BESA can reduce the effects of volume conduction. Actually such sentences does not make sense. The correct here is to talk about forward problem and how the authors deal with it. In any case the main reason of rejecting this work is the large amount of results (Fig. 4, 5, 6, 7, 10) that lead to confusion and making the article very long and difficult to understand. Comments for improvements are to decide which are the most distinctive results, describe them well and consistently in a manner that the authors will attract easily many readers. Take this opportunity and if possible resubmit the work in the same journal.

7. PLOS authors have the option to publish the peer review history of their article (what does this mean?). If published, this will include your full peer review and any attached files.

Reviewer #2: No

- - - - -

---

## [Decision Letter · Decision Letter 2]

3 Oct 2023

PONE-D-22-00971R2A phase-shifting anterior-posterior network organizes global phase relationsPLOS ONE

Dear Dr. Suzuki,

Thank you for submitting your manuscript to PLOS ONE. After careful consideration, we feel that it has merit but does not fully meet PLOS ONE’s publication criteria as it currently stands. Therefore, we invite you to submit a revised version of the manuscript that addresses the points raised during the review process.

The manuscript has been evaluated by four reviewers, and their comments are available below.

The reviewers have raised a number of major concerns. They feel the manuscript should outline a clearly-defined research question, and they request improvements to the reporting of methodological aspects of the study, for example, regarding the exclusion criteria and more information on how the data collection was completed. The reviewers also note concerns about the statistical analyses presented and request re-analyses be completed.

Could you please carefully revise the manuscript to address all comments raised?

We look forward to receiving your revised manuscript.

Kind regards,

Avanti Dey, PhD

Staff Editor

PLOS ONE

Journal Requirements:

Additional Editor Comments (if provided):

Reviewers' comments:

Reviewer's Responses to Questions

**Comments to the Author**

1. If the authors have adequately addressed your comments raised in a previous round of review and you feel that this manuscript is now acceptable for publication, you may indicate that here to bypass the “Comments to the Author” section, enter your conflict of interest statement in the “Confidential to Editor” section, and submit your "Accept" recommendation.

Reviewer #2: (No Response)

Reviewer #3: (No Response)

Reviewer #4: (No Response)

Reviewer #5: (No Response)

2. Is the manuscript technically sound, and do the data support the conclusions?

Reviewer #2: Partly

Reviewer #3: Yes

Reviewer #4: Yes

Reviewer #5: Yes

3. Has the statistical analysis been performed appropriately and rigorously? 

Reviewer #2: I Don't Know

Reviewer #3: Yes

Reviewer #4: Yes

Reviewer #5: Yes

4. Have the authors made all data underlying the findings in their manuscript fully available?

Reviewer #2: Yes

Reviewer #3: Yes

Reviewer #4: Yes

Reviewer #5: Yes

5. Is the manuscript presented in an intelligible fashion and written in standard English?

Reviewer #2: Yes

Reviewer #3: Yes

Reviewer #4: Yes

Reviewer #5: Yes

6. Review Comments to the Author

Reviewer #2: Sorry but the review really disagrees with the authors response, the manuscript by the view of the results / figures really de-motivates any try for real scientific review. It is very difficult to understand and interpret them. Please revise your work making it readable in terms of figure and maybe the reviewer will be willing to really review the work.

Reviewer #3: Manuscript Number: PONE-D-22-00971R2

Manuscript Title: A phase-shifting anterior-posterior network organizes global phase relations

Comments/ Review:

The paper is well written and clear illustrations are provided.

The manuscript presents interesting and novel approach for modelling phase shifting among interiors.

The method and material section is enough detailed to check the suitability skilled investigators.

The paper has ample and both old & latest references are mentioned and should be accepted for the journal.

Reviewer #4: (No Response)

Reviewer #5: In the present manuscript, the authors investigate phase differences between EEG sites during different behavioral conditions: resting with the eyes closed and watching a silent nature video.

They analyze spontaneous fluctuations in phase modularity and compare the spatial distributions of phase relations between states of maximal and minimal modularity.

Even though there are many different ways to preprocess EEG data, their methods are well explained and suitable, especially for the sensor level.

The results are potentially interesting for the computational and experimental neuroscience communities

I would only suggest a more detailed explanation about how they measure the phase modularity instead of only mentioning references 14 and 21. Therefore, I recommend it for publication.

7. PLOS authors have the option to publish the peer review history of their article (what does this mean?). If published, this will include your full peer review and any attached files.

Reviewer #2: No

Reviewer #3: No

Reviewer #4: No

Reviewer #5: **Yes: **Fernanda Selingardi Matias

---

## [Author Response · Author response to Decision Letter 2]

15 Nov 2023

Our response to the specific review comments are attached as a PDF file. Thank you.

---

## [Decision Letter · Decision Letter 3]

20 Dec 2023

A phase-shifting anterior-posterior network organizes global phase relations

PONE-D-22-00971R3

Dear Dr. Suzuki,

We’re pleased to inform you that your manuscript has been judged scientifically suitable for publication and will be formally accepted for publication once it meets all outstanding technical requirements.

Kind regards,

Jianhong Zhou

Staff Editor

PLOS ONE

Additional Editor Comments (optional):

Reviewers' comments:

Reviewer's Responses to Questions

**Comments to the Author**

1. If the authors have adequately addressed your comments raised in a previous round of review and you feel that this manuscript is now acceptable for publication, you may indicate that here to bypass the “Comments to the Author” section, enter your conflict of interest statement in the “Confidential to Editor” section, and submit your "Accept" recommendation.

Reviewer #4: All comments have been addressed

Reviewer #5: All comments have been addressed

2. Is the manuscript technically sound, and do the data support the conclusions?

Reviewer #4: Yes

Reviewer #5: Yes

3. Has the statistical analysis been performed appropriately and rigorously? 

Reviewer #4: Yes

Reviewer #5: Yes

4. Have the authors made all data underlying the findings in their manuscript fully available?

Reviewer #4: Yes

Reviewer #5: Yes

5. Is the manuscript presented in an intelligible fashion and written in standard English?

Reviewer #4: Yes

Reviewer #5: Yes

6. Review Comments to the Author

Reviewer #4: (No Response)

Reviewer #5: The authors have addressed all my comments. Therefore I recommend the paper for publication in Plos One.

7. PLOS authors have the option to publish the peer review history of their article (what does this mean?). If published, this will include your full peer review and any attached files.

Reviewer #4: No

Reviewer #5: No

---

## [Editor Report · Acceptance letter]

1 Feb 2024

PONE-D-22-00971R3 

PLOS ONE

Dear Dr. Suzuki, 

I'm pleased to inform you that your manuscript has been deemed suitable for publication in PLOS ONE. Congratulations! Your manuscript is now being handed over to our production team.

Kind regards, 

on behalf of

Dr. Jianhong Zhou 

Staff Editor

PLOS ONE